# DreamShard: Generalizable Embedding Table Placement for Recommender Systems

**Daochen Zha**[1]    **Louis Feng**[2]    **Qiaoyu Tan**[3]    **Zirui Liu**[1]    **Kwei-Herng Lai**[1]
**Bhargav Bhushanam**[2]    **Yuandong Tian**[2]    **Arun Kejariwal**[2]    **Xia Hu**[1]
[1]Rice University    [2]Meta Platforms, Inc.    [3]Texas A&M University
{daochen.zha,Zirui.Liu,khlai,Xia.Hu}@rice.edu
{lofe,bbhushanam,yuandong,akejariwal}@fb.com
qytan@tamu.edu

## Abstract

We study embedding table placement for distributed recommender systems, which aims to partition and place the tables on multiple hardware devices (e.g., GPUs) to balance the computation and communication costs. Although prior work has explored learning-based approaches for the device placement of computational graphs, embedding table placement remains to be a challenging problem because of 1) the operation fusion of embedding tables, and 2) the generalizability requirement on unseen placement tasks with different numbers of tables and/or devices. To this end, we present DreamShard, a reinforcement learning (RL) approach for embedding table placement. DreamShard achieves the reasoning of operation fusion and generalizability with 1) a cost network to directly predict the costs of the fused operation, and 2) a policy network that is efficiently trained on an estimated Markov decision process (MDP) without real GPU execution, where the states and the rewards are estimated with the cost network. Equipped with sum and max representation reductions, the two networks can directly generalize to any unseen tasks with different numbers of tables and/or devices without fine-tuning. Extensive experiments show that DreamShard substantially outperforms the existing human expert and RNN-based strategies with up to 19% speedup over the strongest baseline on large-scale synthetic tables and our *production* tables. The code is available at https://github.com/daochenzha/dreamshard.

## 1 Introduction

Embedding learning is a commonly used technique to deal with categorical features in deep recommendation models by mapping sparse features into dense vectors [1, 2, 3, 4, 5]. However, the embedding tables can be extremely large due to the large feature sizes [6]. For example, in the YouTube recommendation model, a single categorical feature contains tens of millions of video IDs [7]; the Meta recommendation model demands multi-terabyte memory [8]. Distributed training has been adopted to place the tables on multiple hardware devices such as GPUs [3, 6, 9, 10, 11]. However, even with distributed training, the embedding tables are often still the efficiency bottlenecks. For instance, embedding lookup is shown to dominate the training throughput in the Meta recommendation model [8]. In our internal production model, which has hundreds of tables, embedding lookup accounts for 48% and 65% of the total computation and communication costs, respectively.

How the embedding tables are placed can significantly impact the costs. Figure 1 shows the traces of different placement strategies on a task of placing 50 tables on 4 devices. Typically, embedding lookup consists of four stages. In the forward pass, the sparse indices are mapped into dense vectors (forward computation), which are then sent to the target devices (forward communication).

36th Conference on Neural Information Processing Systems (NeurIPS 2022).

In the backward pass, the gradients of the embedding vectors are sent back from the target devices (backward communication) and applied to the embedding vectors (backward computation). The tables will easily lead to imbalances if not carefully partitioned. The random placement in Figure 1a is bottlenecked by GPU2 with a 56.6 milliseconds latency, while the more balanced placements in Figure 1b and 1c significantly reduce the costs to 42.8 and 35.95 milliseconds, respectively. This work asks: *given a set of embedding tables, how can we identify the best placement of the tables to balance the costs?*

Device placement is essentially a partition problem, which is one of the classical NP-hard combinatorial optimization problems [12]. A recent line of research uses reinforcement learning (RL) for device placement of computational graphs [13, 14, 15, 16, 17, 18, 19, 20]. For example, [13] proposed to train an RNN controller with content-based attention to predict the placement. Other studies advanced [13] in different ways, such as using hierarchical models [14], more sophisticated RL algorithms [15], and graph neural networks [16].

However, embedding table placement remains to be an open and challenging problem due to the operation fusion [21] of tables and the generalizability requirement. **1)** Modern embedding implementations (e.g., FBGEMM [22]), use a single operation to subsume multiple tables for acceleration. The speedup of the fused operation over the sum of the single-table operation costs is not constant and depends on the characteristics of the fused tables (e.g., table dimensions). Our analysis finds that the speedups vary significantly across different table

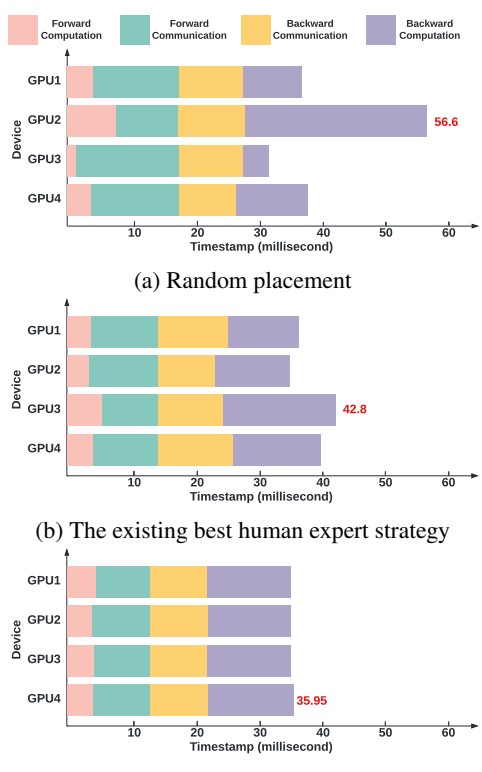

(a) Random placement

(b) The existing best human expert strategy

(c) DreamShard

Figure 1: Visualization of random placement, the existing best human expert strategy, and DreamShard on a task of placing 50 tables on 4 GPUs. The dense computations and communications are omitted in the traces because they do not have an imbalance issue. We provide more visualizations in Appendix L.

combinations, ranging from 1X to 3X (Figure 12 in Appendix A.3.2). Thus, we not only need to reason about cost balance but also how the tables should be fused to maximize the speedup. **2)** In real-world scenarios, the adopted embedding tables and the available devices can change frequently (e.g., machine learning engineers/researchers may conduct concurrent experiments with various table combinations and numbers of devices). Thus, a practical algorithm should generalize to tasks with unseen tables, different numbers of tables, and different numbers of devices. It is non-trivial to achieve this with the existing device placement approaches.

To this end, we introduce DreamShard, an RL approach for embedding table placement. DreamShard achieves the reasoning of operation fusion and generalizability with two novel ideas. **1)** It learns a cost network to directly predict the costs of the fused operations. Specifically, the network takes as input the table features (e.g., table dimension) of each single-table and outputs the computation and communication costs. **2)** It trains a policy network by interacting with an estimated Markov decision process (MDP) without real GPU execution, where the states and the rewards are estimated by the predictions of the cost network. Equipped with sum reductions for the table representations and max reductions for the device representations, the two networks can directly generalize to unseen placement tasks with different numbers of tables and/or devices without fine-tuning.

Extensive experiments show that DreamShard outperforms the existing human expert and RNN-based [13] strategies on open-sourced synthetic tables [23] and our *production* tables, achieving up to 19% speedup over the strongest baseline. Moreover, it can generalize to unseen tasks that have different numbers of tables and/or devices with neglectable performance drop (< 0.5 milliseconds). Additionally, its inference is very efficient. It can place hundreds of tables in less than one second.

## 2 Generalizable Embedding Table Placement Problem

The embedding table placement problem seeks a device placement[1] of all the tables such that the overall cost (in terms of execution time) is minimized (we provide a background for the distributed training of recommendation models in Appendix A.1). Consider $M$ embedding tables $\{\mathbf{e}_1, \mathbf{e}_2, ..., \mathbf{e}_M\}$ and $D$ devices, where $\mathbf{e}_i \in \mathbb{R}^N$ denotes the table features that characterize the embedding lookup patterns. In our work, we use 21 table features, including hash size, dimension, table size, pooling factor, and distribution (their definitions are provided in Appendix A.2). A placement $\mathbf{a} = [a_1, a_2, ..., a_M]$, where $a_i \in \{1, 2, ..., D\}$, assigns each table to a device. Let $c(\mathbf{a})$ denote the cost measured on GPUs. The goal of embedding table placement is to find the $\mathbf{a}$ such that $c(\mathbf{a})$ is minimized. Due to the NP-hardness of the partition problem [12], identifying the exact solution demands extensive computational overhead. Thus, the state-of-the-art algorithms often approximate the optimal partition via sampling with RL [24, 13]. However, sampling remains expensive because obtaining $c(\mathbf{a})$ requires running operations on GPUs. Given that the embedding tables and the available devices can frequently change, we wish to approximate the best $\mathbf{a}$ without GPU execution.

Motivated by this, we study the *generalizable embedding table placement* (GETP) problem. Let $\mathcal{E}$ be the space of all the embedding tables. A placement task can be denoted as $T_i = (\mathcal{E}_i, D_i)$, where $\mathcal{E}_i \subseteq \mathcal{E}$ is a set of tables, and $D_i$ is the number of devices. Given $N_{\text{train}}$ training tasks $\mathcal{T}_{\text{train}} = \{T_1, T_2, ..., T_{N_{\text{train}}}\}$, and $N_{\text{test}}$ testing tasks $\mathcal{T}_{\text{test}} = \{T_1, T_2, ..., T_{N_{\text{test}}}\}$, the goal is to train a placement policy based on $\mathcal{T}_{\text{train}}$ (GPU execution is allowed during training) such that the learned policy can minimize the costs for the tasks in $\mathcal{T}_{\text{test}}$ without GPU execution.

## 3 DreamShard Framework

We present DreamShard, an RL framework based on estimated MDP, to tackle the GETP problem. An overview of the framework is shown in Figure 2. The key idea is to formulate the table placement process as an MDP (Section 3.1) and train a cost network to estimate its states and rewards (Section 3.2). A policy network with a tailored generalizable network architecture is trained by efficiently interacting with the estimated MDP (Section 3.3). The two networks are updated iteratively to improve the state/reward estimation and the placement policy.

Figure 2: DreamShard framework. The agent interacts with the estimated MDP, which is trained with the cost data collected from GPUs.

### 3.1 MDP Formulation

Given embedding tables $\{\mathbf{e}_1, \mathbf{e}_2, ..., \mathbf{e}_M\}$ and $D$ devices, we aim to generate a placement $\mathbf{a} = [a_1, a_2, ..., a_M]$. The key idea is to place the tables one by one at each step, where the state characterizes the tables that have been placed so far, the action is the device ID, and the reward represents the execution time on GPUs. Specifically, at a step $t$, the state $s_t = \{s_{t,d}\}_{d=1}^D$ is all the table features of the tables placed on all the devices, where $s_{t,d} = \{\mathbf{e}_i | i \in \mathcal{P}_d\}$ denotes all the table features corresponding to device $d$ ($\mathcal{P}_d$ is the set of table IDs that have been placed on device $d$). We further augment the raw features with cost features which are obtained by collecting the operation computation and communication times from GPUs (Appendix A.3 provides a comprehensive analysis of the cost features). Formally, the augmented state is defined as $\widetilde{s}_t = \{s_t, \{\mathbf{q}_{t,d}\}_{d=1}^D\}$, where $\mathbf{q}_{t,d} \in \mathbb{R}^3$ has three elements representing forward computation time, backward computation time, and backward communication time for the current operation in device $d$ (we provide detailed explanations of why forward communication time is excluded in Appendix A.4). We find that the augmented cost features can significantly boost the performance, evidenced by the ablations in Table 3. The action $a_t \in \mathcal{A}_t$ is an integer specifying the device ID, where $\mathcal{A}_t$ is the set of legal actions at step $t$. A device ID is considered legal if placing the current table on the corresponding device does not cause a memory explosion. The reward $r_t$ is 0 for all the intermediate steps, and the reward at the final step $M$ is the negative of the cost, i.e., $r_M = -c(\mathbf{a})$, which encourages the agent to achieve lower cost.

---

[1] In this work, we focus on GPU devices, where all the GPU devices are identical, which is the most common configuration in our production. We defer the mixed scenarios of both GPUs and CPUs to future work.

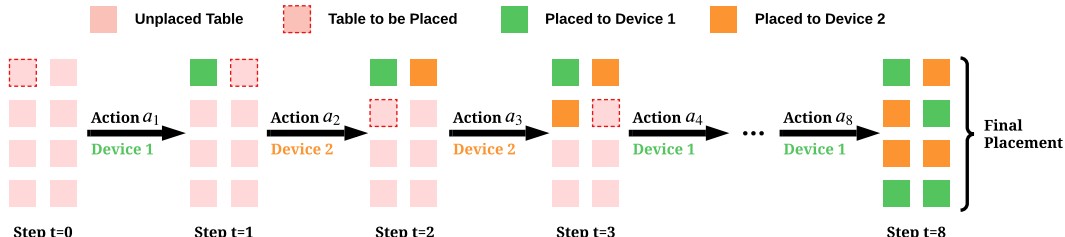

Figure 3: MDP formulation of embedding table placement.

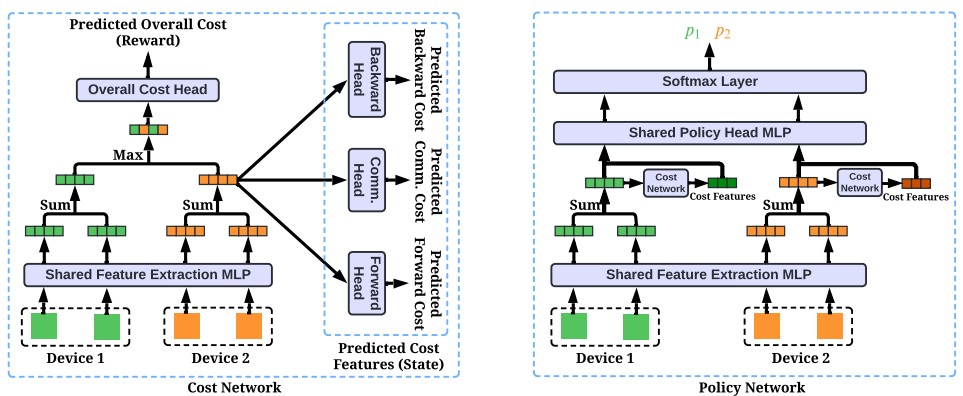

Figure 4: DreamShard's cost network (**left**) and policy network (**right**).

The procedure is illustrated for an example task of placing 8 tables $\{\mathbf{e}_1, \mathbf{e}_2, ..., \mathbf{e}_8\}$ on 2 devices in Figure 3. At step 0, no table has been placed so $s_0 = \{\{\}, \{\}\}$ and the augmented state $\widetilde{s}_0 = \{s_0, \{\mathbf{q}_{0,0}, \mathbf{q}_{0,1}\}\}$, where both $\mathbf{q}_{0,0}$ and $\mathbf{q}_{0,1}$ are zero vectors (i.e., $[0,0,0]$) since all the computation and communication times are 0 as well. Then the action $a_0 = 1$ makes the MDP transit to the next state $s_1 = \{\{\mathbf{e}_1\}, \{\}\}$ with its corresponding augmented state $\widetilde{s}_1 = \{s_1, \{\mathbf{q}_{1,0}, \mathbf{q}_{1,1}\}\}$, where $\mathbf{q}_{1,0}$ becomes a non-zero vector containing the computation and communication costs by running $\{\mathbf{e}_1\}$ and $\{\}$ on GPUs. We repeat the above process, and finally at step 8, we have $s_8 = \{\{\mathbf{e}_1, \mathbf{e}_4, \mathbf{e}_7, \mathbf{e}_8\}, \{\mathbf{e}_2, \mathbf{e}_3, \mathbf{e}_5, \mathbf{e}_6\}\}$. The corresponding $\mathbf{q}_{8,1}$, and $\mathbf{q}_{8,2}$ are the measured times of running $\{\mathbf{e}_1, \mathbf{e}_4, \mathbf{e}_7, \mathbf{e}_8\}$ and $\{\mathbf{e}_2, \mathbf{e}_3, \mathbf{e}_5, \mathbf{e}_6\}$ on two devices. The action sequence $\mathbf{a} = [a_1, a_2, ..., a_8]$ is the generated placement, which is then evaluated on GPUs to obtain the reward.

**Discussion 1.** The MDP enjoys two desirable properties. **1)** The legal action $\mathcal{A}_t$ can guarantee that the generated placement satisfies the memory constraints. **2)** The one-by-one placement enables the agent to be generalized across different numbers of tables. For example, an agent trained on an MDP with very few tables can be applied to another MDP with more tables by simply executing more steps.

**Discussion 2.** A straightforward idea to solve the MDP is to greedily place the current table on the device with the lowest cost at each step, where the cost function can be one of or a combination of the state features (e.g., the sum of the table dimensions, or the sum of all the cost features). However, greedy heuristics are often sub-optimal. Thus, we seek a learning-based algorithm to explore various placement possibilities and make comprehensive decisions based on all the state features.

## 3.2 Learning an Estimated MDP

Interacting with the above MDP is computationally expensive since obtaining the cost features and the reward requires GPU execution. Motivated by world models [25, 26], we build an estimated MDP by approximating the cost features and the reward with a cost network. Let $f_{\text{cost}}$ denote the cost network. $f_{\text{cost}}$ takes as input the raw table features $s_t$, and predicts cost features $\{\mathbf{q}_{t,d}\}_{d=1}^{D}$ and the overall cost $c(\mathbf{a})$. $f_{\text{cost}}$ is trained with mean squared error (MSE) loss using the cost data collected from the GPUs. Once trained, it can predict the cost features or the reward with a single forward pass without GPU execution. However, it is non-trivial to design the architecture of $f_{\text{cost}}$ because it needs

---
**Algorithm 1** Training of DreamShard
---
1: **Input:** Training tasks $\mathcal{T}_{\text{train}}$, the number of data collection steps $N_{\text{collect}}$, the number of cost network update steps $N_{\text{cost}}$, the batch size for updating the cost network $N_{\text{batch}}$, the number RL update steps $N_{\text{RL}}$, the number of considered episodes in each RL update $N_{\text{episode}}$
2: Initialize a cost network, a policy network, and a buffer
3: **for** iteration = 1, 2, ... until convergence **do**
4:     **for** step = 1, 2, ... $N_{\text{collect}}$ **do**              ▷ *Collect cost data from GPUs*
5:         Randomly sample a training task from $\mathcal{T}_{\text{train}}$
6:         Generate a placement by interacting with the estimated MDP using the policy network
7:         Evaluate the placement on the hardware and store the collected cost data to the buffer
8:     **end for**
9:     **for** step = 1, 2, ... $N_{\text{cost}}$ **do**         ▷ *Update the cost network (no GPU execution)*
10:         Randomly sample $N_{\text{batch}}$ cost data from the buffer
11:         Update the cost network based on MSE loss (Eq. 1 in Appendix B.4.1)
12:     **end for**
13:     **for** step = 1, 2, ... $N_{\text{RL}}$ **do**         ▷ *Update the policy network (no GPU execution)*
14:         Randomly sample a training task from $\mathcal{T}_{\text{train}}$
15:         Collect $N_{\text{episode}}$ episodes by interacting with the estimated MDP using the policy network
16:         Update the policy network based on the policy gradient loss (Eq. 2 in Appendix B.4.1)
17:     **end for**
18: **end for**
---

to accommodate different numbers of devices (i.e., $s_t$ can have variable sizes), and different numbers of tables in each device (i.e., $s_{t,d}$ can have variable lengths).

The left-hand side of Figure 4 shows DreamShard's generalizable design of $f_{\text{cost}}$, which is based on two key ideas. **First,** it uses a shared MLP to map raw table features into table representations. For any unseen tables, this MLP can be directly applied to extract table representations. **Second,** it enables a fixed-dimension representation for each device with sum reductions (i.e., the element-wise sum of the table representations in the device), and similarly for the overall representation across devices with max reductions (Appendix B.3 compares different reduction choices and finds that this sum-max combination leads to the most accurate prediction). The reduced representations are then followed by multiple MLP heads for cost predictions. For unseen tasks with different numbers of tables and/or devices, the reductions will always lead to fixed-dimension device/overall representations, so that the prediction heads can be directly applied. Appendix B.1 provides more details.

### 3.3 Training the Policy Network on the Estimated MDP

**Generalizable policy network architecture.** Let $\pi$ be the policy network. $\pi$ maps the augmented state $\widetilde{s}_t = \{s_t, \{\mathbf{q}_{t,d}\}_{d=1}^{D}\}$ to action $a_t$, i.e., $a_t = \pi(\widetilde{s}_t)$. $\pi$ also adopts a generalizable design, shown in the right-hand side of Figure 4. Like $f_{\text{cost}}$, $\pi$ uses a shared MLP and sum reductions to produce a fixed-dimension representation, which is then concatenated with the cost features to obtain the device representation. To accommodate the potentially variable action space (i.e., the number of available devices may vary), a shared MLP will process each device representation separately to obtain a confidence score, followed by a Softmax layer to produce action probabilities. This design allows $\pi$ to generalize across different numbers of devices. Appendix B.2 provides more details.

**Training and inference.** Algorithm 1 summarizes the training procedure of DreamShard, which iteratively executes the following: **1)** collect cost data from GPUs based on the placements generated by the current policy, **2)** update the cost network with the previously collected cost data, and **3)** update the policy network by interacting with the current estimated MDP. Throughout the training process, the estimated MDP gradually becomes more accurate, and the resultant policy network tends to generate better placements. Appendix B.4.2 provides more details of the training procedure. For the inference, the trained cost network and policy network can be directly applied to unseen tasks to generate placements without GPU execution, which is summarized by Algorithm 2 in Appendix B.4.3.

# 4 Experiments

Our experiments aim to answer the following research questions. **RQ1:** How does DreamShard compare with the existing human expert and RL-based placement strategies? **RQ2:** Can DreamShard generalize to placement tasks with different numbers of tables and/or devices? **RQ3:** How efficient is the training of DreamShard? **RQ4:** How do the hyperparameters influence the performance of DreamShard? **RQ5:** How does each component of DreamShard contribute to the performance? **RQ6:** How accurate is the estimated MDP and to what extend can it accelerate the training and inference?

## 4.1 Experimental Setup

**Datasets.** Academic recommendation datasets are often too small to enable a meaningful evaluation because the cost will always be very small no matter how the tables are placed. Thus, we use two industrial-scale datasets. **DLRM**[2] is a large-scale synthetic dataset with 856 tables, recently released by Meta. It shares memory access reuse patterns similar to those arising in Meta production workloads. **Prod** is an internal large-scale dataset for production recommendation models. It has a similar scale as DLRM. The main difference is that DLRM only has a fixed dimension for all the tables, while Prod is more challenging with diverse table dimensions, ranging from 4 to 768. For reproducibility, we mainly focus on the DLRM dataset since it is open-sourced. We only report the main results on the Prod dataset for verification purposes. We provide more details in Appendix C.

**Baselines.** We compare DreamShard against human expert strategies from previous work [27, 8, 28], including **size-based**, **dim-based**, **lookup-based**, **size-lookup-based** greedy balancing strategies. We also include an **RNN-based** RL algorithm [13], which uses RNN architecture to map operators to devices. Since the feature extraction layers of the RNN-based method were designed for other operations instead of embedding tables, for a fair comparison, we adapt [13] by making it have the same feature extraction layers as in DreamShard. We provide more details in Appendix D.

**Configurations.** To evaluate the generalizability of DreamShard, we randomly divide the tables into a training pool $\mathcal{E}^{\text{train}}$ and a testing pool $\mathcal{E}^{\text{test}}$. The two pools have the same number of tables but they are not overlapped. A sharding task $T_i$ is constructed by randomly sampling a subset of $|\mathcal{E}_i|$ tables from a pool, where the number of tables $|\mathcal{E}_i| \in \{10, 20, 30, 40, 50, 60, 70, 80, 90, 100\}$ for the DLRM dataset, and $|\mathcal{E}_i| \in \{20, 40, 80\}$ for the Prod dataset. For all the experiments, we randomly sample 50 training and 50 testing tasks from $\mathcal{E}^{\text{train}}$ and $\mathcal{E}^{\text{test}}$, respectively. DreamShard is trained on the training tasks and will be evaluated on unseen tables in the testing tasks. We denote placement tasks with different numbers of tables and devices using the format of `dataset-num_tables (num_devices)`. For example, DLRM-30 (4) suggests that there are 30 tables sampled from the DLRM dataset in each training/testing task with 4 available devices. We provide more details in Appendix E.

**Implementation Details.** We use the same hyperparameters for all the experiments with $N_{\text{collect}} = 10$, $N_{\text{cost}} = 300$, $N_{\text{batch}} = 64$, $N_{\text{RL}} = 10$, $N_{\text{episode}} = 10$, 10 training iterations, and an entropy weight of 0.001 in the policy gradient. 2080 Ti GPUs and V100 GPUs are used for the DLRM (except that we use V100 for experiments with 8 GPUs) and Prod datasets, respectively. All the experiments are run 5 times, and we report the mean and the standard deviation. Appendix B provides more details.

## 4.2 Results and Analysis

**Evaluation of DreamShard against baselines (RQ1).** We perform qualitative and quantitative comparisons of DreamShard against the baselines. **Qualitatively,** Figure 1 visualizes the traces of DreamShard and the baselines on one of the tasks from DLRM-50 (4). DreamShard achieves significant better overall cost than the best baseline (35.95 vs. 42.8) with **1)** a better balance of forward and backward commutation workloads, and **2)** less communication time, possibly due to a better balance of table dimensions. **Quantitatively,** Table 1 presents comprehensive evaluations on tasks with different numbers of tables and devices on the DLRM and the Prod datasets. **Observations: 1)** DreamShard outperforms the baselines on *all* the tasks. **2)** DreamShard shows strong generalizability on unseen tables, achieving the same level of performance on *all* the testing and training tasks. **3)** DreamShard appears to be more advantageous on harder tasks. Specifically, DreamShard achieves more improvement over the baselines on tasks with more tables/devices on the Prod dataset. In particular, DreamShard achieves 19% improvment over the strongest baseline on Prod-80 (8). **4)**

---

[2]`https://github.com/facebookresearch/dlrm_datasets`

Table 1: Overall cost comparison in milliseconds and relative speedups over random placement, averaged over 50 different randomly sampled tasks. The placement tasks are denoted as `dataset-num_tables` (`num_devices`). For example, DLRM-30 (4) suggests that 30 tables are sampled from the DLRM dataset for each task with 4 GPUs. More results are provided in Appendix F.

| Task | | No strategy | Human Experts | | | | RL | |
|---|---|---|---|---|---|---|---|---|
| | | Random | Size-based | Dim-based | Lookup-based | Size-lookup-based | RNN-based | DreamShard |
| DLRM-20 (4) | Train | 24.0±0.6 | 22.7±0.0 (+5.7%) | 21.3±0.0 (+12.7%) | 19.1±0.0 (+25.7%) | 19.1±0.0 (+25.7%) | 22.4±0.5 (+7.1%) | **18.6±0.2 (+29.0%)** |
| | Test | 23.0±0.5 | 21.7±0.0 (+6.0%) | 19.9±0.0 (+15.6%) | 18.3±0.0 (+25.7%) | 18.4±0.0 (+25.0%) | 20.9±0.3 (+10.0%) | **17.6±0.2 (+30.7%)** |
| DLRM-40 (4) | Train | 41.3±0.2 | 39.6±0.0 (+4.3%) | 37.4±0.1 (+10.4%) | 33.6±0.0 (+22.9%) | 33.6±0.1 (+22.9%) | 39.2±0.7 (+5.4%) | **32.8±0.3 (+25.9%)** |
| | Test | 41.1±0.5 | 40.3±0.0 (+2.0%) | 37.3±0.0 (+10.2%) | 33.0±0.1 (+24.5%) | 33.2±0.0 (+23.8%) | 39.2±1.1 (+4.8%) | **32.4±0.3 (+26.9%)** |
| DLRM-60 (4) | Train | 57.7±0.8 | 56.6±0.1 (+1.9%) | 52.9±0.0 (+9.1%) | 49.2±0.1 (+17.3%) | 49.3±0.0 (+17.0%) | 55.5±0.9 (+4.0%) | **47.6±0.4 (+21.2%)** |
| | Test | 58.1±0.6 | 59.6±0.1 (-2.5%) | 53.7±0.0 (+8.2%) | 48.7±0.2 (+19.3%) | 49.1±0.1 (+18.3%) | 56.0±0.7 (+3.8%) | **47.9±0.7 (+21.3%)** |
| DLRM-80 (4) | Train | 75.7±1.0 | 76.0±0.0 (-0.4%) | 70.0±0.3 (+8.1%) | 64.8±0.0 (+16.8%) | 65.3±0.1 (+15.9%) | 73.2±2.7 (+3.4%) | **62.2±0.2 (+21.7%)** |
| | Test | 74.5±0.8 | 77.7±0.2 (-4.1%) | 69.9±0.4 (+6.6%) | 64.1±0.2 (+16.2%) | 65.1±0.0 (+14.4%) | 72.9±2.4 (+2.2%) | **62.7±0.3 (+18.8%)** |
| DLRM-100 (4) | Train | 91.8±1.7 | 94.1±0.3 (-2.4%) | 86.7±0.3 (+5.9%) | 81.2±0.4 (+13.1%) | 82.2±0.2 (+11.7%) | 94.5±10.7 (-2.9%) | **78.4±0.6 (+17.1%)** |
| | Test | 94.5±6.5 | 95.4±0.0 (-0.9%) | 84.7±0.4 (+11.6%) | 79.5±0.3 (+18.9%) | 80.8±0.3 (+17.0%) | 94.8±13.0 (-0.3%) | **77.8±0.8 (+21.5%)** |
| DLRM-40 (8) | Train | 15.6±0.4 | 14.1±0.0 (+10.6%) | 13.4±0.1 (+16.4%) | **9.8±0.0 (+59.2%)** | 9.9±0.0 (+57.6%) | 16.2±0.8 (-3.7%) | **9.8±0.6 (+59.2%)** |
| | Test | 15.2±0.2 | 14.5±0.0 (+4.8%) | 13.2±0.0 (+15.2%) | 9.5±0.0 (+60.0%) | 9.5±0.0 (+60.0%) | 16.0±1.1 (-5.0%) | **9.4±0.5 (+61.7%)** |
| DLRM-80 (8) | Train | 25.0±0.2 | 24.0±0.0 (+4.2%) | 21.7±0.0 (+15.2%) | 17.1±0.0 (+46.2%) | 17.5±0.0 (+42.9%) | 51.4±3.9 (-51.4%) | **16.1±0.3 (+55.3%)** |
| | Test | 25.2±1.3 | 25.6±0.5 (-1.6%) | 20.8±0.0 (+21.2%) | 16.7±0.2 (+50.9%) | 16.9±0.1 (+49.1%) | 53.4±4.6 (-52.8%) | **16.1±0.4 (+56.5%)** |
| DLRM-120 (8) | Train | 34.0±0.3 | 32.3±0.0 (+5.3%) | 29.8±0.0 (+14.1%) | 24.5±0.0 (+38.8%) | 25.3±0.0 (+34.4%) | 58.6±2.7 (-42.0%) | **23.3±0.2 (+45.9%)** |
| | Test | 33.5±0.5 | 35.0±0.0 (-4.3%) | 29.2±0.0 (+14.7%) | 23.7±0.0 (+41.4%) | 24.5±0.0 (+36.7%) | 58.7±3.1 (-42.9%) | **22.8±0.2 (+46.9%)** |
| DLRM-160 (8) | Train | 42.8±0.3 | 41.6±0.0 (+2.9%) | 39.0±0.0 (+9.7%) | 32.0±0.0 (+33.7%) | 32.7±0.0 (+30.9%) | 58.3±3.5 (-26.6%) | **30.3±0.2 (+41.3%)** |
| | Test | 41.1±0.0 | 42.4±0.0 (-3.1%) | 36.4±0.0 (+12.9%) | 30.8±0.0 (+33.4%) | 31.6±0.0 (+30.1%) | 59.3±5.4 (-30.7%) | **29.6±0.2 (+38.9%)** |
| DLRM-200 (8) | Train | 51.5±1.2 | 48.2±0.0 (+6.8%) | 48.0±0.0 (+7.3%) | 38.9±0.0 (+32.4%) | 39.9±0.0 (+29.1%) | 68.7±2.4 (-25.0%) | **37.2±0.2 (+38.4%)** |
| | Test | 50.7±0.2 | 50.8±0.0 (-0.2%) | 44.8±0.0 (+13.2%) | 38.0±0.0 (+33.4%) | 38.6±0.0 (+31.3%) | 70.4±2.8 (-28.0%) | **36.4±0.3 (+39.3%)** |
| Prod-20 (2) | Train | 41.3±0.7 | 43.4±0.0 (-4.8%) | 37.0±0.0 (+11.6%) | 44.2±0.0 (-6.6%) | 45.8±0.0 (-9.8%) | 38.0±0.3 (+8.7%) | **36.3±0.3 (+13.8%)** |
| | Test | 42.8±0.4 | 46.1±0.0 (-7.2%) | 39.5±0.0 (+8.4%) | 45.9±0.0 (-6.8%) | 45.7±0.0 (-6.3%) | 39.3±0.6 (+8.9%) | **37.5±0.2 (+14.1%)** |
| Prod-40 (4) | Train | 35.1±0.3 | 39.4±0.0 (-10.9%) | 31.3±0.0 (+12.1%) | 36.4±0.0 (-3.6%) | 38.8±0.0 (-9.5%) | 33.9±2.5 (+3.5%) | **28.3±0.3 (+24.0%)** |
| | Test | 38.3±0.3 | 43.6±0.0 (-12.2%) | 33.5±0.0 (+14.3%) | 37.4±0.0 (+2.4%) | 40.1±0.0 (-4.5%) | 36.7±2.8 (+4.4%) | **30.4±0.7 (+26.0%)** |
| Prod-80 (8) | Train | 43.2±0.2 | 44.3±0.0 (-2.5%) | 39.0±0.0 (+10.8%) | 43.7±0.0 (-1.1%) | 49.3±0.0 (-12.4%) | 56.6±6.8 (-23.7%) | **33.6±0.9 (+28.6%)** |
| | Test | 47.7±0.4 | 53.9±0.0 (-11.5%) | 41.9±0.0 (+13.8%) | 46.1±0.0 (+3.5%) | 49.6±0.0 (-3.8%) | 62.5±4.2 (-23.7%) | **35.2±0.8 (+35.5%)** |

Table 2: Generalization performance of DreamShard on target tasks w.r.t. to different numbers of tables (**top**) and devices (**bottom**), averaged over 50 randomly sampled tasks. Appendix G provides the results of more challenging tasks, such as mixed table-wise and device-wise transferring scenarios.

| Source Task → Target Task | Random | Best baseline strategy | DreamShard (trained on target task) | DreamShard (trained on source task) |
|---|---|---|---|---|
| DLRM-20 (4) → DLRM-100 (4) | 94.5±6.5 | 79.5±0.3 (+18.9%) | **77.8±0.8 (+21.5%)** | 77.9±0.4 (+21.3%) |
| DLRM-40 (4) → DLRM-80 (4) | 74.5±0.8 | 64.1±0.2 (+16.2%) | **62.7±0.5 (+18.8%)** | 62.7±0.5 (+18.8%) |
| DLRM-80 (4) → DLRM-40 (4) | 41.1±0.5 | 33.0±0.1 (+24.5%) | **32.4±0.3 (+26.9%)** | 32.4±0.2 (+26.9%) |
| DLRM-100 (4) → DLRM-20 (4) | 23.0±0.5 | 18.3±0.0 (+25.7%) | **17.6±0.2 (+30.7%)** | 17.7±0.3 (+29.9%) |
| DLRM-20 (4) → DLRM-20 (2) | 29.9±0.4 | 26.0±0.0 (+15.0%) | **25.8±0.2 (+15.9%)** | **25.8±0.1 (+15.9%)** |
| DLRM-40 (4) → DLRM-40 (2) | 58.6±0.7 | 52.4±0.0 (+11.8%) | **51.9±0.1 (+12.9%)** | 52.0±0.3 (+12.7%) |
| DLRM-20 (2) → DLRM-20 (4) | 23.0±0.5 | 18.3±0.0 (+25.7%) | **17.6±0.2 (+30.7%)** | 17.8±0.3 (+29.2%) |
| DLRM-40 (2) → DLRM-40 (4) | 41.1±0.5 | 33.0±0.1 (+24.5%) | **32.4±0.3 (+26.9%)** | 32.6±0.3 (+26.1%) |

RNN-based method is only better than the random strategy on tasks with few tables/devices, but is worse on harder tasks. A possible reason is that RNN-based algorithm does not have a cost network, and using RL alone could lead to unstable performance. **5)** Lookup-based strategy is the best baseline on the DLRM dataset, while dim-based strategy is better on the Prod dataset. A potential reason is that the tables in the Prob dataset have very diverse table dimensions, while the tables in the DLRM dataset have the same dimension. As such, the dimensions on Prod tasks can more easily become imbalanced, leading to poor communication efficiency. Dim-based strategy can better balance the dimensions, which leads to a better overall performance. DreamShard outperforms the baselines on both the DLRM and the Prod datasets, showing its flexibility in dealing with different scenarios.

**Analysis of generalizability (RQ2).** In Table 2, we directly apply a DreamShard model trained from one task to another task *without any fine-tuning* (the rightmost column), where the source and the target tasks have different numbers of tables and/or devices. DreamShard shows neglectable performance drop, suggesting that it is generalizable across different numbers of tables and/or devices.

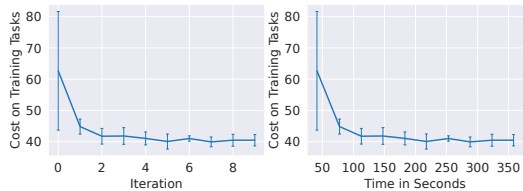

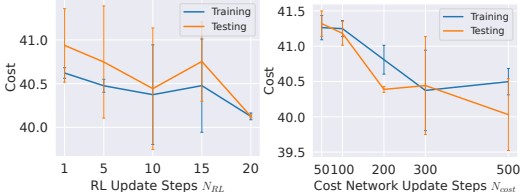

Figure 5: Performance (↓) of DreamShard on the DLRM-50 (4) datasets w.r.t. the numbers of iterations (**left**) and running time (**right**). Curves for other tasks are in Appendix H.

Figure 6: Impacts of the number of RL update steps $N_{RL}$ (**left**) and cost network update steps $N_{\text{cost}}$ (**right**) on the DLRM-50 (4) dataset. Curves for other tasks are in Appendix I.

Table 3: Ablation study of DreamShard. Results on other tasks are in Appendix J.

| Task | | w/o dim | w/o hash size | w/o pooling factor | w/o table size | w/o distribution | w/o cost | w/ RNN | DreamShard |
|---|---|---|---|---|---|---|---|---|---|
| DLRM-50 (4) | Train | 40.8±0.4 | 40.7±0.1 | 46.3±0.3 | 40.8±0.4 | 40.6±0.2 | 47.5±1.2 | 40.5±0.2 | **40.4±0.5** |
| | Test | 40.9±0.6 | 40.6±0.3 | 47.2±0.1 | 40.6±0.7 | 40.5±0.2 | 46.3±0.1 | 40.5±0.1 | **40.4±0.6** |

**Analysis of training efficiency (RQ3).** Figure 5 plots the performance of DreamShard w.r.t. the number of iterations and running time in seconds on four 1080Ti GPUs. The training of DreamShard is highly efficient. On DLRM-50 (4), it can achieve strong performance in less than 5 iterations or 200 seconds. Note that we only need to re-train or fine-tune DreamShard when the table pools have significant changes. Once trained, it only needs a forward pass for inference.

**Hyperparameter study (RQ4).** We study the impacts of two key hyperparameters: **1)** $N_{\text{RL}}$, which controls the RL update frequency, and **2)** $N_{\text{cost}}$, which determines the cost network update frequency. We vary one of them with the other fixed, shown in Figure 6. Increasing $N_{\text{RL}}$ or $N_{\text{cost}}$ will both lead to improvement, suggesting that both the cost network and the policy network need to be sufficiently trained. However, when $N_{\text{RL}}$ and $N_{\text{cost}}$ are large enough, increasing them will not bring more improvement. Considering that larger values will lead to more computational costs, we set $N_{\text{RL}} = 10$ or $N_{\text{cost}} = 300$ as a trade-off between the performance and the training efficiency.

**Ablation study (RQ5).** We study the importance of each table feature and check whether the RNN architecture helps, with the following ablations. **1)** We remove each of the features in the state. **2)** We add an RNN upon the device representation in the policy network. We makes several observations from the results in Table 3. **1)** Cost features play a significant role, which demonstrates the effectiveness of our proposed augmented state. **2)** The most contributing table features are the pooling factor and the dimension, which aligns with our intuitions since these two feature are the determining factors of computation and communication workloads. **3)** Using more features leads to consistently good performance. **4)** While the policy makes decisions sequentially, RNN does not provide clear benefits. This is why we have kept the architecture simple with only MLP in DreamShard.

**Study of the estimated MDP (RQ6). First,** we study how many data points are required to train an accurate cost network, and how accurate the cost network needs to be to enable a strong policy. Specifically, we randomly sample 10,000 cost data points from the DLRM-50 (4) dataset. Then we use 20% for testing, and vary the size of the training data to train the cost network. Further, we fully train a policy network with 100 iterations based on each of the trained cost networks. We make two observations from the results in Figure 7. **1)** As expected, more data points lead to a more accurate cost network. **2)** Interestingly, after around 100 data points, the policy network does not keep improving even though the cost network becomes more accurate. Thus, we only need a sufficiently (but not perfectly) accurate cost estimation to achieve the best performance. This also partially explains why DreamShard can generalize: even though the cost network could be not very accurate on unseen tables, DreamShard can still find strong placements with the policy network. **Second,** we study the necessity of the estimated MDP. We consider a variant that obtains the cost features and rewards directly from GPUs, shown in Figure 8. Using the estimated MDP can make the training and the inference orders of magnitudes faster, while achieving the same level of performance. In particular, the inference time is less than one second even with a hundred tables.

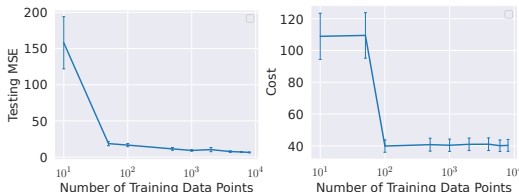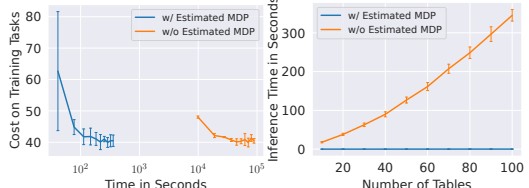

Figure 7: MSE of the cost network w.r.t. the number of training data points collected from GPUs (**left**), and the corresponding performance of a fully trained policy network based on the trained cost network (**right**) on the DLRM-50 (4) dataset.

Figure 8: Training curves w/ or w/o the estimated MDP on DLRM-50 (4) dataset (**left**), and their inference times w.r.t. the number of tables (**right**). Using cost network to estimate the MDP leads orders of magnitudes faster training and inference.

## 5 Related Work

**Embedding tables.** Embedding tables are commonly used to deal with sparse features in recommendation models [1, 2, 3, 4, 5, 29, 30, 31]. However, the extremely large embedding tables are often the storage and efficiency bottlenecks [6, 7, 8, 3, 6, 9, 10, 11, 32]. To our knowledge, the only two studies that target the embedding table placement problem are RecShard [27] and our previous work AutoShard [33]. RecShard approaches table placement at a per-row granularity through exploitation of the underlying feature distributions. RecShard leverages these distributions, along with characteristics of the training system, to shard embedding tables across a tiered memory hierarchy using a mixed integer linear program, with more frequent rows placed in GPU HBM and the remaining placed in CPU DRAM. In contrast, our work develops a neural cost network for cost estimation and an RL-based optimization algorithm, and focuses on sharding across a single memory layer as opposed to a tiered memory hierarchy. AutoShard also leverages RL for embedding table sharding. However, it only balances the computational costs, which will lead to sub-optimal solutions since communication also account for significant costs. In addition, our DreamShard is more efficient in training than AutoShard due to the design of the estimated MDP. Another line of work focuses on reducing the embedding table sizes [34, 35, 36, 37, 38, 39, 40], which is orthogonal to our work since DreamShard is also applicable to compressed tables.

**Device placement optimization.** The existing device placement techniques can be mainly grouped into two categories: RL-based algorithms, and cost modeling methods. **1)** RL-based algorithms treat the device placement as a black box and optimize the cost objective in a trial-and-error fashion [13, 14, 15, 16, 17, 18, 19, 20]. Unfortunately, these methods are computationally expensive and often require training from scratch on unseen tasks. While Placeto [16] shows generalizabily on computational graphs with graph embeddings, it can not deal with various table combinations since there is no graph structure, and it can not handle different numbers of devices. **2)** Cost modeling methods build a cost model to reflect the real performance and adopt offline algorithms (e.g., scheduling, and dynamic programming) to optimize the placement [41, 42, 43, 44, 45]. However, the cost model could be inaccurate. In particular, they can not deal with the operation fusion of embedding tables. Whereas, DreamShard combines the advances of RL with a neural cost model for accurate cost prediction.

**Deep RL.** Deep RL has recently made significant progress in games [46, 47, 48, 49, 50, 51, 52, 53, 54]. Our work is related to using RL to optimize machine learning model designs, such as neural architecture search [55, 56, 57, 58, 59], data augmentation [60], data sampling [61, 62], pipeline search [63, 64, 65]. However, these methods often only focus on one task and can not generalize to unseen tasks. Our work is related to meta-learning [66, 67, 68]. Instead of performing meta-learning for machine learning tasks, we focus on machine learning system design. Our work is also related to solving combinational optimization problems with RL [24, 69]. Unlike the above studies, we show that RL can tackle a practical problem of embedding table placement and the learned policy is generalizable.

## 6 Conclusions and Future Work

We present DreamShard for embedding table placement in recommender systems. We formulate the problem as an MDP which places the tables one by one at each step. Then we leverage RL to

solve the MDP. To accelerate the training and the inference, we build an estimated MDP by training a cost network to approximate the state features (i.e., computation and communication times) and the reward (i.e., the overall cost), leading to orders of magnitudes faster training and inference speeds. Extensive experiments on the open-sourced DLRM dataset and our production dataset demonstrate the superiority of DreamShard over the existing algorithms. Moreover, DreamShard shows strong generalizability, making it a desirable choice in real-world applications. In the future, we will extend DreamShard to tiered memory hierarchy and large-scale training clusters with complex topologies.

## Acknowledgements

The work is, in part, supported by NSF (#IIS-2224843). The views and conclusions in this paper are those of the authors and should not be interpreted as representing any funding agencies. We would also like to thank the helpful feedback from the anonymous reviewers.

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
