# A  Background of Embedding Table Placement

In this section, we provide a background of *embedding table placement* problem (also called *embedding table sharding* [27, 8] since it essentially partitions the tables across different devices). In Section A.1, we introduce a background of distributed training of recommendation models. Section A.2 lists some important table features, which characterize the table accessing patterns and are highly related to computation/communication costs. Section A.3 further provides an in-depth analysis of the correlation between the computation/communication costs and the table features. Finally, we discuss the difference between forward and backward communication times in Section A.4.

## A.1  Distributed Training of Recommendation Models and Embedding Tables

Industrial recommendation models often require massive memory consumption and high training throughput. Thus, distributed training solutions have been developed to train recommendation models. While various recommendation models have been developed in the past decades, they often rely on embedding tables to map sparse categorical features to dense vectors [8, 9, 7, 70, 71, 72]. We take DLRM [3] as an example to introduce distributed training design since DLRM is the core of the official package of PyTorch for recommendation models[3] and is commonly used in both academia and industry.

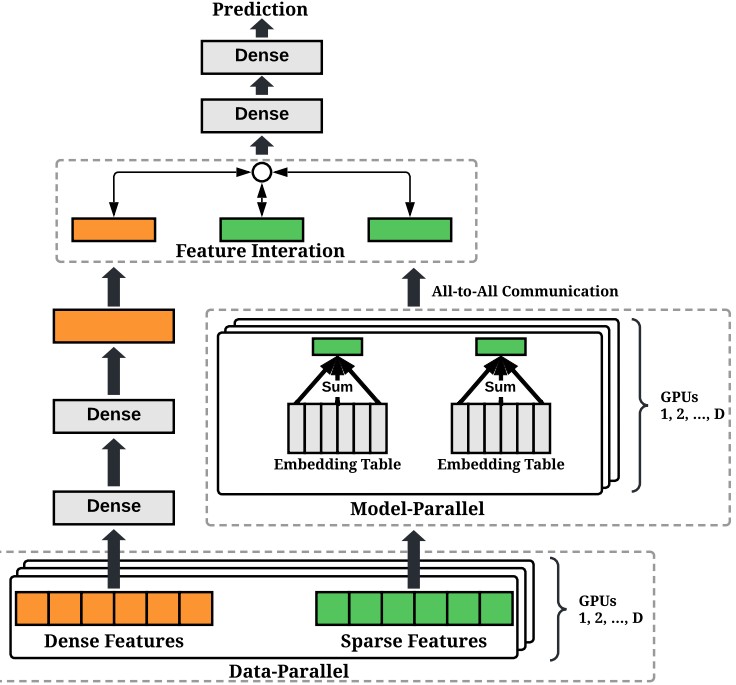

Figure 9: A typical distributed training framework for recommendation models [3]. The figure is adapted from [33].

Figure 9 shows an overview of the DLRM model. DLRM processes two types of features, i.e., dense features, and sparse features. Dense features are numerical values and are directly processed with MLPs in DLRM. Sparse features are categorical features. For example, in the context of YouTube video recommendation, a possible sparse feature can be video IDs. For the sparse features, DLRM adopts embedding tables to map the categorical features to dense vectors. Specifically, each row of an embedding table corresponds to a feature value (i.e., video ID), and the number of columns corresponds to the vector dimension. Given a list of feature values, an embedding table lookup is performed to obtain the vectors. For each feature value, a corresponding vector is obtained from the table via hashing. Then all the obtained vectors are summed to obtain a fixed-dimension vector. The embedding lookup is performed for all the tables. The obtained vectors are processed by MLPs, and then will be interacted with the dense representations to obtain the final representation. The

---

[3]https://github.com/pytorch/torchrec

final MLP will be processed by another MLP, which maps the representation to the predictions (e.g., click-through rate).

However, in real-world applications, the embedding tables can become extremely large and can not be fed into a single GPU. Meanwhile, the dataset can also be extremely large so using one GPU may not meet the high training throughput requirement. To accommodate the massive memory and training throughput requirements, DLRM adopts a combination of data-parallelism and model-parallelism. For the data-parallelism, DLRM replicates MLPs on each device and partitions training data into different devices. In this way, each device only needs to process its own mini-batch of data, achieving higher training throughput. For the model-parallelism, the embedding tables are placed on different devices. With this design, in the forward pass, the embedding lookup for a certain feature value will be performed by querying the device that actually holds the corresponding table. For example, suppose a feature value in the training data of device 1 corresponds to table 1. If table 1 is unfortunately placed in device 2, then device 1 will query device 2 to obtain the vector via communication. The above communication is essentially very frequent if we feed a batch of data for training. Thus, in actual implementations, such communication is often batched by sending a batch of data at a time (termed all-to-all communication since there is often communication between each pair of the devices). In the backward pass, the accumulated gradients will be similarly sent back to the device that actually holds the table. In the above example, device 1 will calculate the gradient and send the gradient tensor back to device 2 to update the embedding table.

We summarize the overall training procedure of DLRM as follows. In the forward pass, each device samples its own mini-batch of data, which contains a batch of dense features and a batch of sparse features. The dense features will be simply processed by the duplicated MLP to obtain dense representations. The sparse features (i.e., feature values, or the indices of embedding tables) will be sent to the corresponding devices for embedding lookup. Then each device will perform the embedding lookup for the tables that are placed on the device (**forward computation**). The obtained vectors are sent back to the device that launches the query, which is essentially an all-to-all communication since each device will communicate with all the other devices (**forward communication**). The obtained vectors will be interacted with the dense vectors to obtain a final representation, followed by a prediction head to make the predictions. In the backward pass, the gradient will be passed backwardly from the prediction loss. Updating the dense part is straightforward since it is the same as the standard backpropagation. For the sparse counterpart, the gradient of each vector needs to be sent back to the device that actually holds the corresponding table, which leads to another all-to-all communication (**backward communication**). Then, the gradient will be applied to the embedding tables to update the embedding weights (**backward computation**). At the end of the backward pass, the weights of the duplicated MLP will be synchronized.

We only focus on optimizing the cost of the sparse part of the model, i.e., the cost of embedding tables, including forward computation, forward communication, backward communication, backward computation. This is because the costs of embedding computation and communication often dominate the overall training efficiency. For example, in our internal training pipeline of production recommendation models (which is already well optimized with numerous iterations), the cost of embedding tables account for 48% and 65% of the total computation and communication costs, respectively. Meanwhile, the embedding table cost is orthogonal to other costs, such as data loading, dense feature processing, etc. This means embedding table cost optimization can be considered as an independent task, which will contribute to the overall training throughput. We note that the embedding computation and communication can be performed simultaneously with the computation of the dense MLP. The bottleneck depends on which part takes more time. However, we observe in production models that embedding cost is often significantly larger than the dense MLP cost due to the extremely large embedding tables, which aligns with the observations from previous studies [8, 6]. Thus, the dense MLP cost is often "hidden" by the embedding table cost, and embedding table cost becomes the bottleneck during model training.

Optimizing the embedding table cost is very challenging because it has very complex computation and communication patterns. **First,** embedding computation or communication alone has very complex relationship with the embedding lookup patterns (we will provide detailed quantitative analysis of this in Section A.3. **Second,** the forward/backward computation/communication costs can have interactive effects. For example, if the forward communication of a device is significantly larger than those of the other devices. Then the other devices need to wait until this device finishes the forward computation so that they can obtain the queried embedding vectors. Similarly, the backward

computation for a device can only start after the device receives all the gradients in the backward communication.

As a result, different embedding table placements will significantly impact the embedding cost in several aspects. **First,** a good combination of embedding tables may lead to faster forward/backward computation since it may enable a more efficient kernel implementation. **Second,** balancing the forward computation time can reduce the waiting time before the forward communication starts. **Third,** balancing the backward computation time will also reduce the waiting time. **Forth,** balancing the number of amount of data being sent can accelerate the all-to-all communication.

However, optimizing embedding table placement is very challenging. This partition problem is known to be NP-hard[4], which means the number of possible placements grows exponentially with more tables. Additionally, due to the complexity of the embedding table cost discussed above, it is hard to optimize the placement in an analytical way. This motivates our work of DreamShard, which leverages RL to optimize the embedding table placement in a trial-and-error fashion.

## A.2 Embedding Table Features

We define several embedding table features to characterize embedding tables. These table features are highly correlated to the computation and communication workloads. Thus, in DreamShard, they serve as the input of the cost network and the policy network. In total, we use 21 features, which are defined as follows.

- **Dimension (dim, 1 feature)**: It is the dimension of each embedding vector, i.e., the number of columns of the embedding table. It is a critical table feature since it determines the workloads of both computation and communication. For computation, the forward pass requires fetching the embedding vectors, and the backward pass will apply gradients to the embedding vectors, both of which have a computational complexity that increases linearly with the vector dimension. For communication, the vector dimension determines the data size, which will impact the communication time.

- **Hash size (1 feature)**: It is the number of embedding vectors in the embedding table, i.e., the number of rows of the table. It is called hash size because embedding lookup is essentially a hashing operation. While hashing is often believed to have $O(1)$ time complexity, which means the lookup time does not depend on the hash size, we find that hash size can still impact the lookup time because of caching mechanism. Specifically, modern GPUs often have L1/L2 caches, which are small but faster memories. If hash size is small, a larger portion of the embedding vectors can be put into the caches such that the lookup will be faster. In contrast, a large hash size will lead to a smaller portion of the embedding vectors being cached such that the lookup time will be larger.

- **Pooling factor (1 feature)**: It is the number of embedding indices in a lookup. For example, in YouTube video recommendation, a user may have watched multiple videos in the past. If a feature corresponds to "all the videos that were watched in the past", then we need to fetch all the embedding vectors that correspond to these video IDs from the table. In this context, pooling factor refers to the number of video IDs. Like dimension, pooling factor decides the workload of computation. In the forward pass, a larger pooling factor will result in more embedding vectors being fetched and summed, which will naturally lead to more computation. Similarly, in the backward pass, more computation will be required to update the embedding vectors with the gradients. Note that pooling factor usually will not impact communication since it does not decide the data size in communication. Since we often adopt mini-batch training, which means a batch of indices will be used to perform embedding lookup, we use the mean pooling factor as the table feature. Specifically, for a batch of indices, we calculate mean value of the pooling factors of all the training samples in the batch.

- **Table size (1 feature)**: Table size is the memory consumption of the embedding table in GBs. It can help the agent reason about satisfying the memory constraints of devices.

- **Distribution (17 feature)**: It refers to the accessing frequencies of all the indices of a table. Specifically, certain indices can be accessed far more frequently than other indices.

---

[4] https://en.wikipedia.org/wiki/Partition_problem

Modern embedding table implementation will exploit such patterns with caching. The indices that are frequently accessed will tend to be put into the L1/L2 cache for acceleration. For a batch of $65,536$ indices, we use 17 bins, including $(0, 1]$, $(1, 2]$, $(2, 4]$, $(4, 8]$, $(8, 16]$, $(16, 32]$, $(32, 64]$, $(64, 128]$, $(128, 256]$, $(256, 512]$, $(512, 1024]$, $(1024, 2048]$, $(2048, 4096]$, $(4096, 8192]$, $(8192, 16384]$, $(16384, 32768]$, and $(32768, \infty)$. We count the number of appearances of each index and assign the count to the corresponding bin. Finally, we normalize the counts and make them sum to 1, which leads to a probability distribution with 17 table feature values.

## A.3 Quantitative Analysis of Computation and Communication Times

Embedding table placement is a very challenging problem because it is hard to estimate the costs without running the operations on GPUs. The main challenges include the non-linear relationship between the table cost and table features, operation fusion, and complex communication patterns. Here, we provide a quantitative analysis of these phenomena. All the results are collected using a modern embedding bag implementation from FBGEMM[5] [22] from 2080Ti GPUs. Note that the results in Section A.3.1 and Section A.3.2 are originally collected in [33].

### A.3.1 Relationship Between Table Cost and Table Features

Recall that in Section A.2, we have defined some table features, which can quantify the workloads of computation. However, due to the parallelism of GPUs, the actual table cost has a non-linear relationship with the table features. Here, we study the relationship between single-table cost and dimension, hash size, pooling factor, and distribution (table size is excluded because it can be essentially inferred from dimension and hash size). Dimension and hash size describe the table itself since they define the numbers of rows and columns of the table, respectively. The pooling factor and distribution characterize the indices assessing patterns, where the pooling factor measures the overall workload, and the distribution features measure the sparsity of the indices distributions. Now we analyze the above two types of features separately with synthetic embedding tables and indices.

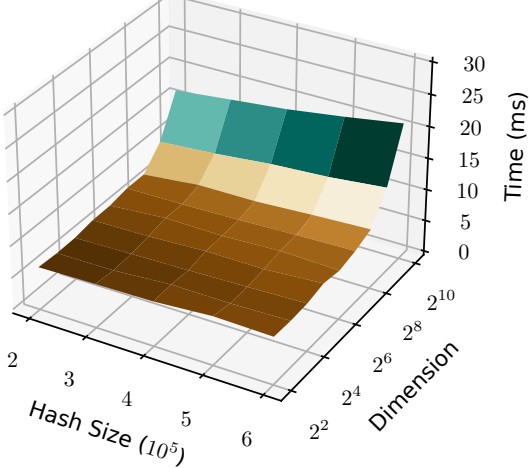

Figure 10: Influence of the hash size and the dimension on single-table cost. The results are originally collected from [33].

We study the impact of dimension and hash size with the pooling factor fixed as 32 and indices to be uniformly distributed. We vary the hash size from $2 \times 10^5$ to $6 \times 10^5$ and dimension from $2^2$ to $2^{10}$. We measure the kernel time (the sum of the forward and backward computation times) of the embedding operation for each of the combinations of hash size and dimension. The heat map of embedding cost is shown in Figure 10. We make three observations. **First,** a higher dimension will significantly increase the kernel time. This is expected since the embedding dimension corresponds to the size of the data to be fetched in the forward pass and the size of the data to be updated with the gradients in the backward pass. **Second,** while hash size only has a moderate impact on the table cost, a large hash size leads to a higher table cost. This also aligns with our intuition since a larger hash

---

[5]https://github.com/pytorch/FBGEMM/

size will lead to a smaller portion of the indices being cached. **Third,** we find that the table cost has a non-linear relationship with both dimension and hash size.

Next, we study the impact of pooling factor and indices distributions with the hash size fixed as $10^6$ and dimension fixed as $32$. We vary the mean pooling factor from $2^0$ to $2^8$. For the indices distribution, some indices could be accessed far more frequently than others [8]. We simulate this phenomenon in our synthetic indices by only allowing a subset of all the embedding vectors to be accessed. Specifically, we define *accessed indices ratio* as the ratio of the embedding vectors that can be accessed in the embedding table. For example, a ratio of 1.0 suggests the indices are uniformly distributed. A ratio of $10^{-3}$ means only $0.1\%$ of all the embedding vectors can be accessed. This means that those $0.1\%$ of embedding vectors are "warm" vectors and can be accelerated with caching. Note that there can be many ways to simulate the indices distributions. Here, we only focus on the most simple one, which masks a subset of the embedding vectors. The impacts of the pooling factor and accessed indices ratio are illustrated in Figure 11. We make three observations. **First,** a larger pooling factor will significantly increase the table cost. This is because a larger pooling factor suggests more computation cost of fetching and updating the embedding vectors. **Second,** sparser indices distribution tends to have lower table cost, which could be explained by the caching mechanism. **Third,** the table cost has a complex and non-linear relationship with pooling factor and indices distributions.

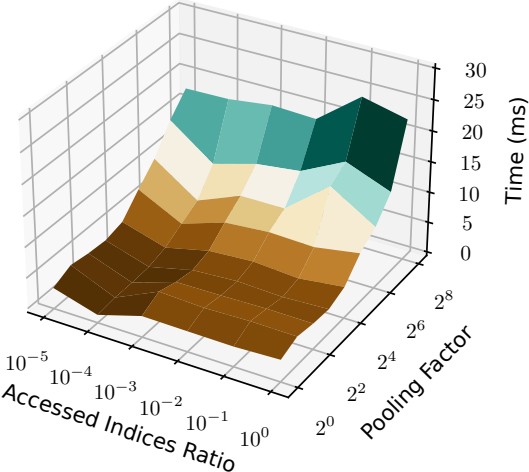

Figure 11: Influence of the indices distribution and the pooling factor on single-table cost. The results are originally collected from [33].

In the above analysis, we separately studied two table features with the other features fixed. However, it is possible that these features have an interaction effect, which could make the table cost even more challenging to estimate. Thus, when developing DreamShard, we are motivated to use a cost network to directly predict the table cost in a data-driven manner.

### A.3.2 Analysis of Operation Fusion

Operation fusion [21] is a common acceleration strategy that uses a single operation to subsume the computation performed by multiple operations. It is particularly effective for embedding tables due to batching. It can often lead to significant speedup in operation computation time. Unfortunately, the operation fusion also makes the multi-table costs hard to predict. Here, we analyze the operation fusion by randomly sampling 10 tables from the DLRM dataset and comparing its multi-table cost and the sum of the single-table costs. We consider the sum of the single-table costs as the baseline because it represents the case without any acceleration. We repeat the sampling 50 times and plot the results in Figure 12.

We make two observations as follows. **First,** the multi-table cost is significantly lower than that of single-table cost. This is expected since operation fusion can accelerate the operation. The results show that operation fusion can lead to roughly 1.5X speedup when we have 10 tables. **Second,** while the multi-table cost is in general positively correlated with the sum of single-table costs, they are not linearly correlated. Specifically, the actual speedup is case-by-case, which may depend on many

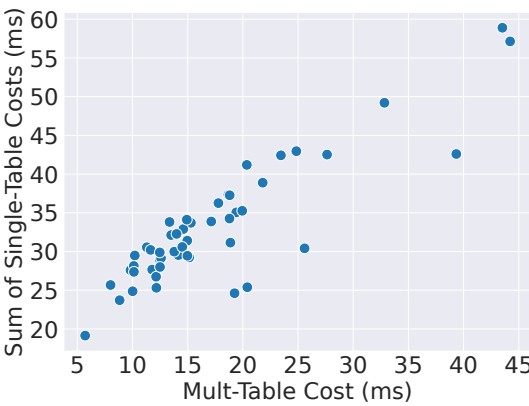

Figure 12: Mult-table cost = the sum of single-table costs? Mult-table cost is smaller than the sum of single-table costs, where the speedup ranges from 1X to 3X. They do not have a clear linear correlation. Assuming that the multi-table cost has a linear correlation with the sum of single-table costs and the single table cost estimation is perfect, we grid-search the correlation efficiency in the range of $[1.0, 2.0]$ with a step size of 0.001. The best MSE we found is 77.97, which is significantly worse than our cost network (our cost network can have less than 1.0 MSE). The results are originally collected from [33].

factors. The results suggest that simply using the sum of single-table costs to estimate the multi-table cost is inaccurate.

In our embedding placement process, we inevitably need to estimate the multi-table costs. Unfortunately, the above analysis suggests that we may not be able to get an accurate estimation without actually running the multi-table operations on GPUs. This motivates us to develop a neural cost network to directly approximate the multi-table costs.

### A.3.3   Analysis of Communication

Embedding tables in recommendation models have very complex communication patterns because of combined model-parallelization and data-parallelization. Specifically, we require all-to-all communication to send the embedding vectors or gradients from device to device. Since there are often limited bandwidths among GPUs, if the data are not distributed in a balanced way, then it may take significantly more time for communication. Here, we analyze the communication costs with different degrees of balance.

The communication cost depends on the amount of data to be sent to each device. In the context of embedding table placement, we mainly need to send the summed embedding vectors, whose sizes are determined by the batch size and the table dimension. Since batch size is pre-determined, the communication cost is essentially decided by the sum of table dimensions in each GPU device. Thus, in our empirical analysis, we adjust the sums of table dimensions for the GPUs to simulate different levels of imbalance. Specifically, we fix the batch size to be $65,536$ and construct 16 embedding tables, where each table has a dimension of $64$. Then we place these tables on the GPU devices to simulate different degrees of balance.

Table 4 presents the communication costs under different degrees of balance using 4 GPUs. We can see that when the sums of table dimensions become more imbalanced, the communication cost also increases. Thus, we need to balance the table dimensions to minimize communication costs. However, balancing dimension alone cannot achieve an overall good result, since a placement with a balanced dimension may still be not balanced in computation. To tackle this challenge, DreamShard jointly optimizes both computation and communication in a data-driven manner with RL.

### A.4   Why Backward Communication Time but not Forward Communication Time?

Forward communication and backward communication will send data of the same size in all-to-all communication (but in different directions). Specifically, in the forward pass, the obtained sparse representations will be sent, while in the backward pass, the gradient of the sparse representations will

Table 4: Communication time of 4 GPUs with different sums of table dimensions (the tables have $1,024$ dimensions in total) in each device with a batch size of $65,536$. We highlight the max communication time across devices since the slowest device becomes the bottleneck.

| Category | Sum of Table dimensions | | | | Communication cost | | | | Max cost |
|---|---|---|---|---|---|---|---|---|---|
| | GPU 1 | GPU 2 | GPU 3 | GPU 4 | GPU 1 | GPU 2 | GPU 3 | GPU 4 | |
| Perfectly Balanced | 256 | 256 | 256 | 256 | 11.24±0.17 | 11.15±0.12 | 11.08±0.17 | 11.08±0.17 | 11.24±0.17 |
| Slightly Imbalanced | 192 | 256 | 320 | 384 | 13.20±0.16 | 13.88±0.04 | 14.18±0.04 | 14.15±0.06 | 14.15±0.06 |
| | 192 | 192 | 320 | 320 | 11.74±0.08 | 13.01±0.13 | 12.89±0.10 | 12.93±0.11 | 13.01±0.13 |
| | 128 | 192 | 320 | 384 | 12.28±0.06 | 13.82±0.08 | 14.03±0.08 | 14.02±0.07 | 14.03±0.08 |
| | 128 | 128 | 384 | 384 | 12.02±0.10 | 14.67±0.09 | 14.73±0.08 | 14.47±0.11 | 14.73±0.08 |
| Very Imbalanced | 64 | 128 | 384 | 448 | 12.91±0.80 | 16.00±0.81 | 16.11±0.81 | 15.82±0.08 | 16.11±0.81 |
| | 64 | 64 | 448 | 448 | 12.50±0.05 | 16.65±0.06 | 16.67±0.08 | 16.29±0.06 | 16.67±0.08 |
| | 64 | 64 | 320 | 576 | 12.56±0.13 | 15.61±0.15 | 16.93±0.17 | 16.89±0.11 | 16.93±0.17 |
| | 64 | 64 | 64 | 832 | 13.01±0.14 | 12.96±0.15 | 17.65±0.21 | 17.65±0.22 | 17.65±0.21 |

be sent back. Recall that the cost network in DreamShard only predicts the backward communication cost instead of the forward communication cost. One may ask why they are different given that the amount of data is the same.

We do not predict forward communication because we find that a considerable portion of the forward communication time is not spent on communication, but instead on idle time waiting for other devices. For example, if a device finishes forward computation very quickly, then it has to wait for other devices to finish computation before it can start communication. However, such waiting time is also counted in the forward communication with PyTorch even though it does not communicate. On the contrary, the devices will be "synced" when the forward communication is finished, so that the backward communication often does not have idle time. Thus, we only predict backward communication, which can better reflect the true communication cost.

# B   Implementation Details

In this section, we introduce the implementation details of DreamShard. We will first introduce the neural architectures of the cost network and the policy network. Then we provide more details of the training and inference procedures. Further, we summarize the hyperparameter configurations. Finally, we describe the hardware and software used in our experiments. **To ensure reproducibility, we will open-source our code**.

## B.1   Neural Architecture of cost network

The cost network consists of three sub-networks, including 1) a shared feature extraction MLP (denoted as $\text{MLP}_\text{table}$), which maps the 21 table features to latent representations, 2) backward/communication/forward heads, which predict cost features based on the device representation, and 3) an overall cost head, which takes the final representation for all the devices as input and predicts the overall cost.

We provide the detailed procedure of a forward pass as follows. For a raw state $s_t$, we first use an MLP to process all the raw table features with $\mathbf{h}_i^\text{table} = \text{MLP}_\text{table}(\mathbf{e}_i) \in \mathbb{R}^l$, where $l$ is the hidden dimension, and $\text{MLP}_\text{table}$ is shared across all the tables. This leads to a set of hidden representations for each device $\{\mathbf{h}_i^\text{table} | i \in \mathcal{P}_d\}$. Then we obtain the device representation with element-wise sum by $\mathbf{h}_d^\text{device} = \sum_{i \in \mathcal{P}_d} \mathbf{h}_i^\text{table} \in \mathbb{R}^l$, which has a fixed dimension regardless of the number of tables in the device. The motivation for using element-wise sum is that $\mathbf{h}_i^\text{table}$ is expected to describe the computational cost patterns of a table so it is natural to accumulate $\mathbf{h}_i^\text{table}$ to represent the potentially accumulated computational costs when we have multiple tables. Then $\mathbf{h}_d^\text{device}$ serves as the input of the backward/communication/forward heads to predict the cost features $\mathbf{q}_{t,d}$. To predict the overall cost, we similarly obtain a fixed-dimension representation for all the devices $\mathbf{h}$ by applying an element-wise max to $\mathbf{h}_d^\text{device}$, i.e., $\mathbf{h}$ is defined by $h_k := \max_{1 \le d \le D} h_{d,k}^\text{device}$, where $h_k$ and $h_{d,k}^\text{device}$ denote the $k^{th}$ element of $\mathbf{h}$, and $\mathbf{h}_d^\text{device}$, respectively. The motivation of element-wise max is that the slowest device is usually the bottleneck of the overall cost. Then $\mathbf{h}$ is followed by an overall cost head to predict the reward

We elaborate on the neural architectures of the three sub-networks as follows.

- **Shared table feature extraction MLP:** The dimension of the latent representation is set to be 32. We instantiate the shared feature extraction MLP with a 2-layer neural network with a size of 21-128-32.

- **Backward/communication/forward heads:** We use three MLPs to implement these three heads. Each MLP is a 2-layer neural network with a size of 32-64-1.

- **Overall cost head:** Similarly, we instantiate it with a 2-layer neural network with a size of 32-64-1.

For all the above three sub-networks, we use the ReLU activation function and the default parameter initialization in PyTorch.

## B.2   Neural Architecture of Policy Network

The policy network consists of three sub-networks, including 1) a shared feature extraction MLP that is independent of that of the cost network (denoted as $\widetilde{\text{MLP}}_{\text{table}}$), which maps the 21 table features to latent representations, 2) cost feature MLP, which processes the cost features by mapping them to latent representations, and 3) a policy head, which maps device representations to probability distributions.

We provide the detailed procedure of a forward pass as follows. **First,** following the cost network, another MLP is used to process the raw table features $s_t$ with $\widetilde{\mathbf{h}}_i^{\text{table}} = \widetilde{\text{MLP}}_{\text{table}}(\mathbf{e}_i) \in \mathbb{R}^l$, where $\widetilde{\text{MLP}}_{\text{table}}$ is shared across all the tables (but independent of $\text{MLP}_{\text{table}}$ in $f_{\text{cost}}$). A fixed-dimension device representation for each device can be similarly obtained with element-wise sum by $\widetilde{\mathbf{h}}_d^{\text{device}} = \sum_{i \in \mathcal{P}_d} \widetilde{\mathbf{h}}_i^{\text{table}} \in \mathbb{R}^l$. **Second,** we augment $\widetilde{\mathbf{h}}_d^{\text{device}}$ with the cost features $\mathbf{q}_{t,d}$. Specifically, we use an MLP to process $\mathbf{q}_{t,d}$ by $\mathbf{h}_d^{\text{cost}} = \text{MLP}_{\text{cost}}(\mathbf{q}_{t,d}) \in \mathbb{R}^l$, where $\text{MLP}_{\text{cost}}$ is shared across the cost features of all the devices. The augmented device representation is the concatenation of $\widetilde{\mathbf{h}}_d^{\text{device}}$ and $\mathbf{h}_d^{\text{cost}}$, denoted as $[\widetilde{\mathbf{h}}_d^{\text{device}}; \mathbf{h}_d^{\text{cost}}]$. **Third,** we use a shared policy head to process the augmented device representation, followed by a Softmax layer to produce action probabilities. Let $\text{MLP}_{\text{policy}}$ be the policy head. The probabilities for all the legal actions are obtained by $\mathbf{p} = \text{Softmax}\{\text{MLP}_{\text{policy}}[\widetilde{\mathbf{h}}_d^{\text{device}}; \mathbf{h}_d^{\text{cost}}] | d \in \mathcal{A}_t\}$, where $\text{MLP}_{\text{policy}}$ is shared across all the devices. **Finally,** we sample an action $a_t$ based on the action probabilities $\mathbf{p}$. Our design allows $\pi$ to be trained on one task and generalize to other tasks with different numbers of tables and/or devices.

We elaborate on the neural architectures of three sub-networks as follows.

- **Shared table feature extraction MLP:** It has the same architecture like that of the cost network (but the weights are not shared). The dimension of the latent representation is set to be 32. We instantiate the shared feature extraction MLP with a 2-layer neural network with a size of 21-128-32.

- **Cost feature MLP:** This network maps the three cost features into a 32-dimension cost representation. We instantiate it with an MLP of size 3-64-32.

- **Policy head:** It takes as input the concatenated representation of the device representation and the cost representation. Thus, its input size is 64 (32 for the device representation and 32 for the cost representation). Then we use a 1-layer MLP of size 64-1 to map the representations to a "confidence score". After obtaining the score for each device, we use a Softmax layer to produce the action probabilities, i.e., the probability of selecting each of the devices.

For all the above three sub-networks, we use the ReLU activation function and the default parameter initialization in PyTorch.

## B.3   Comparison of Different Reductions

Recall that we use the element-wise sum to aggregate table representations in a device and element-wise max to aggregate device representations. Here, we justify our design choices by comparing with

other reduction methods. Specifically, we randomly sample 10,000 cost data points from the DLRM-50 (4) dataset. Then we use 20% for testing and vary the size of the training data to compare the performances of different reductions using different numbers of data points. For all the experiments, we use a batch size of 64, an Adam optimizer with a learning rate of 0.0005, and we train 50,000 batches. We report the sum of testing MSE for all the predicted costs. All the experiments are repeated 5 times, and the mean and standard deviation are reported.

In the first experiment, we try max and mean reductions for the table representations and use max reduction for the device representations . The results are reported in Figure 13. We observe that sum reduction is the best choice for table representations. In the second experiment, we try sum and mean reductions for the device representations and use sum reduction for the table representations. The results are reported in Figure 14. We observe that max reduction is the best choice for table representations. Thus, in DreamShard we use sum reduction for the table representations and max reduction for the device representations.

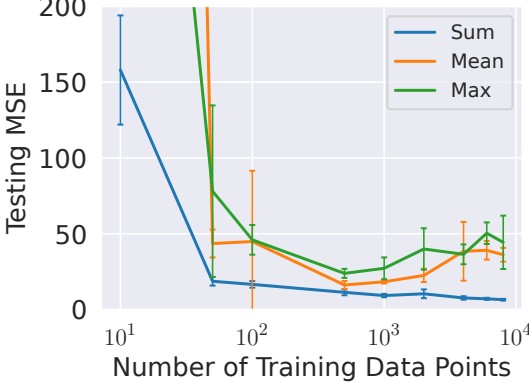

Figure 13: Comparison of different reductions for table representations (max reduction is applied to device representations).

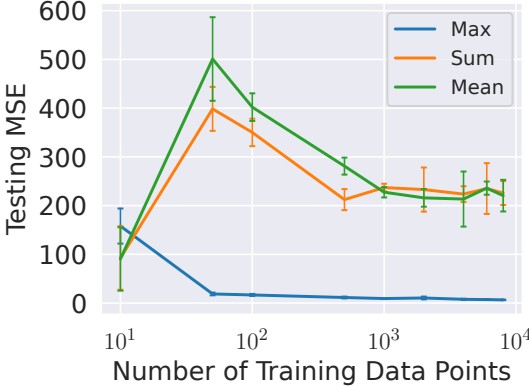

Figure 14: Comparison of different reductions for device representations (sum reduction is applied to table representations).

## B.4    Details of DreamShard Training and Inference

In this subsection, we elaborate on the training and inference procedures. We will first present the loss functions for updating the cost network and the policy network. Then we summarize the training procedure. Finally, we describe how DreamShard performs inference on unseen embedding table placement tasks.

### B.4.1    Loss Functions

We update the cost network with mean squared error (MSE). Recall that we use a buffer to collect cost data by using the current policy $\pi$ to interact with the environment (we will elaborate on this procedure in Section B.4.2). Suppose we have already collected some cost data. Then we use the cost data to update the cost network. Specifically, the cost network maps the raw state $s_t$ into the predicted cost features $\{\hat{\mathbf{q}}_{t,d}\}_{d=1}^{D}$ and the predicted overall cost $\hat{c}$. Let $\{\mathbf{q}_{t,d}\}_{d=1}^{D}$ and $c(\mathbf{a})$ be the

ground truth of the cost features and the overall cost, respectively. We use the sum of their mean squared errors (MSE) to update the cost network:

$$L_{\text{cost}} = \sum_{d=1}^{D} \text{MSE}(\hat{\mathbf{q}}_{t,d} - \mathbf{q}_{t,d}) + \text{MSE}(\hat{c}, c(\mathbf{a})), \tag{1}$$

where $\text{MSE}(\cdot, \cdot)$ represents the MSE loss. Note that it is possible to use a weighted loss to prioritize the prediction of the cost features or the overall cost by introducing additional hyperparameters. We will explore this possibility in our future work.

We update the policy network $\pi$ with the standard policy gradient loss [73] enhanced by a baseline to reduce the variance, and an entropy loss to enhance exploration:

$$L_{\text{RL}} = \sum_{t=0}^{M-1} \log \pi(a_i|s_i)(\sum_{j=i}^{M} r_j - b_j) + w_{\text{entropy}} \sum_{a=1}^{D} \pi(a|s_i) \log \pi(a|s_i), \tag{2}$$

where $\pi(a|s)$ is the predicted probability probability of performing action $a$ in state $s$, $w_{\text{entropy}}$ is the weight of the entropy. $b_j$ is the reward obtained at step $t$; $r_j$ is the negative of the overall cost when $j = M$, and $r_j$ is 0 for all the other steps. Thus, $\sum_{j=i}^{M} r_j$ essentially reduces to $r_M$ (we use $\sum_{j=i}^{M} r_j$ so that it is consistent with the formulas used in the RL literature)[6]. $b_j$ is a baseline to reduce variance and stabilize training. In each update step, we run $N_{\text{episode}}$ episodes at a time and use their mean reward as $b_j$. Then we update the policy $\pi$ by calculating the loss with a batch of episodes. The policy network $\pi$ can be updated with the loss using the standard backpropagation.

### B.4.2 Training Procedure

The training procedure is iterative. In each iteration, we sequentially do the following: 1) use the current policy $\pi$ to sample some placements, collect the costs from GPUs, and store the collected cost data into the buffer, 2) update the cost network with the cost data collected in the buffer, and 3) update the RL agent by interacting with the estimated MDP simulated by the cost network. We provide details for each of the three stages below.

**Data collection.** In this stage, we use the current policy $\pi$ to generate table placements and evaluate the placements on GPUs. Specifically, we first randomly select a training task from $\mathcal{T}_{\text{train}}$. Then we generate a placement for this task by interacting with the estimated MDP with $\pi$. Before starting an episode, we first sort the tables in descending order based on the single-table cost, which is predicted using the cost network. The motivation is that it will be more likely to achieve a good balance if we put the costly table at the beginning of the MDP. Then we follow the MDP to place the tables one by one, where in each step, we obtain the augmented state using the current cost network, and then feed the augmented state to the policy $\pi$ to predict the action probabilities. Then we sample an action based on the action probabilities to make the placement decision. After generating a placement, we evaluate the placement on GPUs to collect the computation and communication costs using PARAM Benchmark [7], which is the official micro-benchmarking tool for PyTorch. To precisely measure the cost, the benchmarking consists of three steps: 1) the initialization step will initialize the operators with the specified embedding table arguments and load the indices data to the GPU, 2) the warmup step will run all the computation and communication for 5 times to allow CUDA to complete the necessary preparations, and 3) the benchmarking step will run all the computations and communications again for 10 times. The median latency in the benchmarking step will be returned since the median value is less sensitive to outliers. The returned latency will be stored in a buffer for training the cost network later. We find that the above benchmarking strategy is very stable, and the obtained latency has very low variance. There is one hyperparameter in this stage, i.e., $N_{\text{collect}}$, which specifies the number of placements to be generated.

**Training the cost network.** In this stage, we sample multiple mini-batches of cost data from the buffer to update the cost network. Specifically, in each update step, we sample a batch of cost data with a size of $N_{\text{batch}}$. Then we feed the data to the cost network and update it based on the loss in

---

[6]In the RL literature, a discount factor is often applied to make the early decisions have a smaller reward. In our context, we simply set the discount factor to be 1 (i.e., no discount) because the reward in the MDP is sparse and the early decisions are very important.

[7]`https://github.com/facebookresearch/param`

Eq. 1. We update it for $N_{\text{cost}}$ times. $N_{\text{batch}}$ and $N_{\text{cost}}$ are hyperparameters controlling the update of the cost network.

**Training the policy network.** We use the current policy $\pi$ to interact with the estimated MDP, which is akin to the data collection stage. The only difference is that we do not evaluate the generated placement on GPUs. Instead, the final reward is simply obtained by a forward pass of the cost network. The design of the estimated MDP can significantly improve the training efficiency of RL since it isolates the RL training from the evaluation on GPUs. In each update step of RL, we first randomly select a training task. Then we generate $N_{\text{episode}}$ episodes through interacting with the estimated MDP. Next, we update the policy network $\pi$ based on Eq. 2. We repeat the above procedure $N_{\text{RL}}$ times.

### B.4.3   Inference Procedure

The inference of DreamShard is straightforward. The procedure is similar to the data collection except that we choose the action with the highest predicted probability instead of sampling an action based on the probabilities. This is because in training, we require the agent to explore different actions and discover the best strategy. Whereas, during inference, we no longer need exploration. As such, we can simply choose the most confident action. We summarize the procedure of performing inference on testing tasks in Algorithm 2. We note that the inference does not require GPUs.

---
**Algorithm 2** Inference of DreamShard
---
1: **Input:** Some testing tasks $\mathcal{T}_{\text{test}}$, the trained cost network, trained policy network
2: **for** each task in $\mathcal{T}_{\text{test}}$ **do**
3:    **for** step = 1, 2, ... until episode ends **do**
4:       Construct the augmented state using the cost network
5:       Predict the action probability using the policy network
6:       Take and record the action that has the highest probability
7:    **end for**
8:    Record the action sequence (placement)
9: **end for**
---

### B.5   Hyperparameter Configuration

We summarize all the hyperparameters of DreamShard below.

- **Data collection:** We set $N_{\text{collect}} = 10$.
- **cost network training:** We set $N_{\text{cost}} = 300$, and $N_{\text{batch}} = 64$.
- **Policy network training:** We set $N_{\text{RL}} = 10$, $N_{\text{episode}} = 10$, and the entropy weight $w_{\text{entropy}} = 0.001$.
- **Optimizer:** For both the cost prediction and policy networks, we adopt Adam optimizer with an initial learning rate of $0.0005$, with the other hyperparameters as default. A linear scheduler is used to linearly decay the learning rate to zero throughout the training process.
- **Embedding operation:** We use the embedding bag implementation in FBGEM[5] [22]. For the parameters of embedding tables, we randomly initialize them with fp16 precision.

### B.6   Hardware and Software Description

For the DLRM dataset, all the experiments are conducted on a server with 48 Intel(R) Xeon(R) Silver 4116 CPU @ 2.10GHz processors, 188 GB memory, and four NVIDIA GeForce RTX 2080 Ti GPUs. For the Prod dataset, the server has similar hardware configurations but with NVIDIA V100 GPUs to accommodate the larger sizes of the tables. For software, we use Python 3.8.4, and PyTorch 1.9.1.

## C   Details of the Datasets

We note that our goal is not to evaluate the accuracy of a recommendation model, but rather the training efficiency of embedding tables. The public recommendation datasets are not suitable for evaluation since they cannot match the scale of real-world industrial models. They are often too small

with very few categorical features so the latency of embedding operations will always be very small no matter how the embedding tables are placed.

Table 5: Comparison of embedding table feature statistics between some popular public recommendation datasets and the industrial-scale DLRM dataset. Our production data has an even larger scale than the DLRM dataset (details are not shown due to data privacy). The public datasets are not suitable to evaluate embedding table placement algorithms. They only have very few small tables, and the average pooling factor is only 1, which suggests there is only one feature in each table per instance when performing embedding lookup.

| Dataset | | # of Tables | Avg. hash size | Avg. pooling factor |
|---|---|---|---|---|
| Public | Criteo[8] | 26 | 17,839 | 1 |
| | Avazu[9] | 23 | 67,152 | 1 |
| | KDD[10] | 10 | 601,908 | 1 |
| Industrial-Scale | DLRM | 856 | 4,107,458 | 15 |

Fortunately, Meta recently released the DLRM[2] dataset, which is a synthetic dataset that shares memory access reuse patterns similar to those arising in Meta production recommendation workloads. This dataset is an ideal benchmark to evaluate embedding table placement algorithms because it can well simulate the real workloads under different table placements in industrial models, and the results obtained on it will be reproducible since the dataset is open-sourced. Table 5 compares the scales of some large-scale public recommendation datasets and the DLRM dataset. We can observe a clear gap between the public datasets and the DLRM dataset. The DLRM dataset has around one order of magnitude more tables, average hash size (i.e., the number of rows of the table), and average pooling factor (i.e., the number of rows extracted in a table for one instance when performing lookup). In what follows, we introduce and visualize the DLRM dataset. We will not provide more details of our production dataset due to data privacy.

## C.1 Data Format of the DLRM Dataset

The DLRM dataset is stored as three PyTorch tensors, which are pickled in a single file. The three tensors include an indices tensor, an offsets tensor, and a length tensor. For brevity, we denote them as `indices`, `offsets`, and `lengths`, respectively. `indices` is a vector, where each element is an integer. The indices are ordered by the keys of (`table_id`, `batch_offset`). For example, the first batch of indices (the size is determined by the offset) is for the first table, and the second batch of indices (the size is determined by another offset) is for the second table, etc. `offsets` is also a vector. It indicates the starting position and the ending position of `indices` for one lookup. It is also ordered by (`table_id`, `batch_offset`). For instance, suppose the batch size is 45. Then `offsets[45]` and `offsets[46]` specify the starting and ending positions of the 45th indices lookup in the first table. The slice between the starting and ending positions, i.e., `indices[offsets[45]:offsets[46]]` corresponds to the 45th instance in the batch for the first table. `lengths` is a matrix and is of the shape of [`num_tables`, `batch_size`], where each element is the pooling factor of the corresponding indices lookup. `lengths` is provided for correctness validation purposes.

## C.2 Data Visualization of the DLRM dataset

We visualize the 856 tables in the DLRM dataset. Specifically, we focus on the distributions of hash size, mean pooling factor, and the relation between the hash size and pooling factor. We also visualize the distribution of indices accessing frequency since it may impact the caching mechanism. Note that all the results are originally collected in [33].

Figure 15 visualizes the distribution of hash size. We observe that the hash sizes for most tables are around $10^6$, while some can reach $10^7$. The tables with large hash sizes could lead to very large tables, making it challenging to balance the size of the tables.

Figure 16 shows the distribution of mean pooling factors. We find that the pooling factor generally follows a power-law distribution. Most of the tables have a pooling less than 5, while there are few tables that have a pooling factor larger than 100 (some certain tables can have a pooling factor of up to 200). Recall that the pooling factor is one of the most important factors that decide the computation workloads. The power-law distribution will make the computation easily imbalanced across devices.

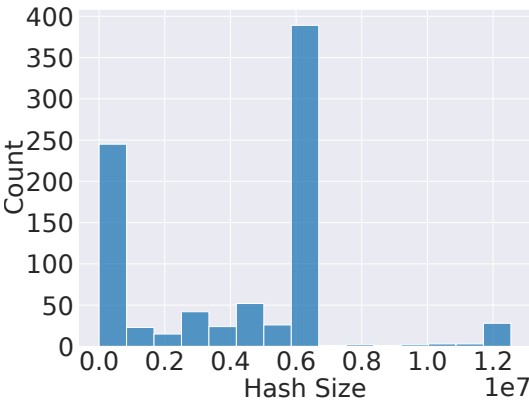

Figure 15: Hash size distribution.

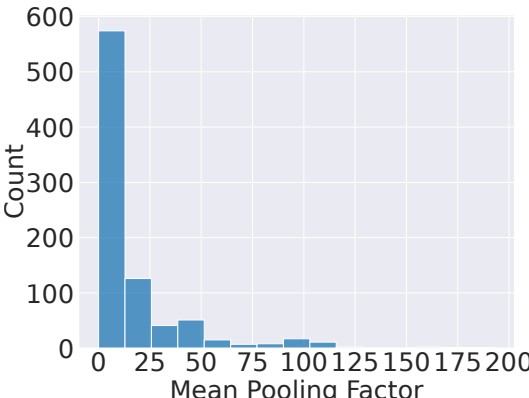

Figure 16: Pooling factor distribution.

We are interested in studying whether the pooling factor and the hash size have a positive correlation. The intuition is that if a table has more values, more rows could be selected when performing embedding lookup. If they have a positive correlation, balancing one of them could also lead to a balance of the other. We plot their relationship in Figure 17. We observe that there is no clear relationship between the hash size and pooling factor. Thus, an ideal algorithm may need to balance both of them to achieve the best results.

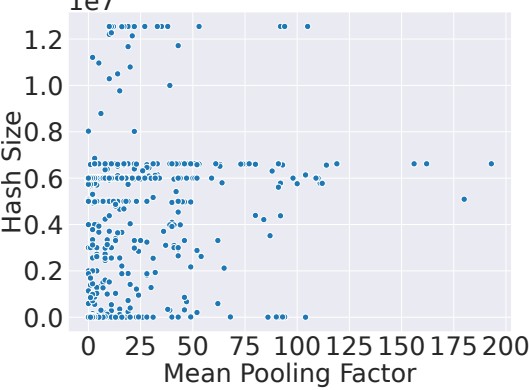

Figure 17: Hash size vs. pooling factor.

Figure 18 illustrates the indices accessing frequency distribution. We observe that most of the indices are accessed less than ten times, while some of them can reach $10^5$. Similarly, the diverse indices accessing frequency will easily lead to imbalances.

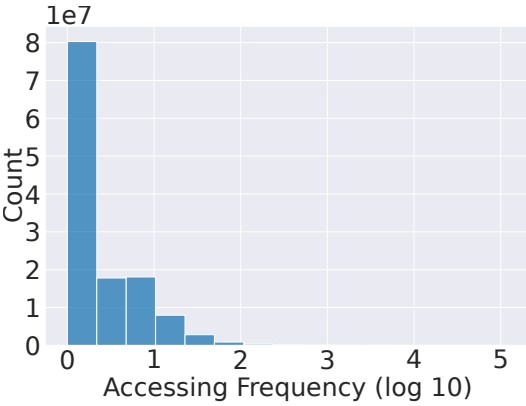

Figure 18: Indices frequency distribution.

Overall, we find that the table features are quite diverse, and can easily lead to imbalances. Specifically, if we do not carefully partition the tables, some tables with high computation costs can be easily put into the same device, resulting in a very high cost for the device. Meanwhile, the imbalances may also lead to heavy communication costs. Thus, an ideal placement algorithm need to globally balance different aspects to achieve the best results. This motivates us to develop learning-based algorithms for embedding table placement.

### C.3 Data Processing

Since the DLRM dataset does not specify the table dimension, we set the table dimension to be 16 for all the tables. We purposely make the dimensions small to facilitate reproducibility on GPUs with small memory. Note that our production dataset in general has a much larger table dimension that is up to 768. In addition, each table can have a different dimension. The large and diverse table dimensions will make the embedding table placement problem more challenging since imbalanced dimensions will significantly and negatively impact both the computation and the communication times. Nevertheless, our results in Table 1 suggest that DreamShard can well deal with the tables that have larger and more diverse dimensions, showing the effectiveness and flexibility of our algorithm.

## D  Details of the Baselines

We compare DreamShard with two types of baselines, including human expert strategies [27, 8, 28], and an RNN-based placement algorithm [13]. We will elaborate on them below.

### D.1  Human Expert Strategies

These strategies have been mentioned or used in previous work of distributed recommender systems [27, 8, 28], and we have adopted them in our internal training workflow for years. The main idea is to use a greedy algorithm to balance the costs, where the costs are estimated based on a specific table feature, or a combination of the table features. These strategies consist of two steps as follows.

- **Cost function:** Each table will be assigned an estimated cost, which serves as the target to be balanced.

- **Greedy algorithm:** The greedy algorithm tries to balance the sum of the costs in each device. Specifically, it first sorts all the embedding tables in descending order based on their costs. In this way, we can more easily achieve a balance if placing the tables greedily. Then starting from the table with the highest cost, we make a greedy decision in each step by placing the current table to the device that has the lowest sum of the cost so far. In the end, each device will roughly have the same or a similar sum of the costs so that we can achieve the goal of load balance.

The various expert strategies mainly differ in how the cost function is designed, i.e., the balancing objective. Specifically, the following cost functions are used as baselines to balance different aspects of the workloads:

- **Size-based:** We use the table size to estimate the cost. The intuition is that the table size is related to both the dimension and the hash size, which can reflect the workloads. In addition, balancing the size can reduce the risk of memory explosion.

- **Dim-based:** We use the table dimension to estimate the cost. Recall that in Section A.3, table dimension can determine both computation and communication workloads. In particular, dimension is the only factor for communication workloads theoretically. Thus, balancing the sums of dimensions is a natural idea.

- **Lookup-based:** We use the product of the table dimension and the pooling factor to estimate the cost. The motivation is that the table dimension and the pooling factor determine the computation workload in lookup.

- **Size-lookup-based:** We use the product of the table dimension, the pooling factor, and the table size to estimate the cost. This is the most comprehensive estimation (but it may not necessarily be the best).

The human expert strategies have several drawbacks. **First,** the estimation could be inaccurate. As shown in Section A.3.1, the actual cost has a non-linear relationship with all the table features and can not be simply approximated with products. **Second,** it only optimizes the sum of the costs and can not model the operation fusion, as analyzed in Section A.3.2. **Third,** while these strategies could achieve good performance in different scenarios, none of them can accommodate all scenarios. For example, if the communication bandwidths are low and communication is the bottleneck, the dim-based strategy could work better. Whereas, if the computation is the bottleneck, the lookup-based strategy may work better. It is difficult to select the most suitable one for real-world applications.

DreadShard addressed all of the above drawbacks with a learning-based cost network and a learning-based placement policy. The cost network directly approximates the multi-table costs in a data-driven manner, which can model the non-linear relationship between the cost and the table features. It can also inherently consider the operation fusion effect since it directly approximates the multi-table costs. Moreover, the RL-based placement policy makes decisions in a data-driven manner so that it can accommodate different scenarios.

## D.2    RNN-based Algorithm

The main motivation for adopting this baseline is that embedding table placement also belongs to general device placement problems. The state-of-the-art algorithms leverage RL to optimize the device placement [13, 15, 16]. Thus, adapting the existing device placement algorithms to the embedding table placement problem is a natural idea. We focus on the RNN-based method proposed in [13] because it is a pioneering work that applies RL to device placement problems, and many of the follow-up studies are motivated by and developed based on this work.

The original RNN-based algorithm uses an RNN controller to sequentially make decisions for device placement, and the RNN controller is updated with the RL loss. **First,** each operation is represented as some operation features, such as data types and output shapes. **Second,** the operation representations are sequentially fed into an RNN architecture. **Third,** an attention layer is applied to the hidden states. **Fourth,** the representation obtained after the attention layer is followed by a policy head to make predictions. **Finally,** the RNN controller will be updated using the standard policy gradient loss.

We have adapted the original RNN-based algorithm so that it can be applied to our embedding table placement problem. Specifically, we replace the operation features with the 21 table features used in DreamShard. Additionally, we use the same feature extraction MLP with the same architecture as DreamShard. The policy head of the RNN-based baseline also has the same architecture as the policy head in DreadShard. The main difference is that we use an RNN and an attention layer to process the feature representations. We note that such design can not generalize across different numbers of devices due to architecture constraints of RNN.

## E   Details of the Experimental Configurations

In this section, we provide more details of how we perform the experiments to test the generalizability of DreamShard. We consider three types of generalizability, including 1) unseen placement tasks (i.e., the combination of the tables is different, but the individual tables may or may not be seen in training), 2) unseen embedding tables, and 3) different numbers of tables/devices. Our experiments are designed to maximally test all these three types of generalizability.

To test 1) and 2), we control the table pools for training and testing. Specifically, we divide all the tables in half to construct a training pool and a testing pool, where the training tasks are constructed only based on the training pool, and the testing tasks are constructed only based on the testing pool. Since there is no overlap of tables between training and testing pools, all the tables in the testing tasks are unseen. To construct each training/testing task, we randomly sample a subset of tables from the corresponding pool, and the number of tables varies from the set $\{10, 20, 30, 40, 50, 60, 70, 80, 90, 100\}$; that is, we consider different combinations of the tables, and we consider the cases from very few tables to many tables. All the table combinations in the testing tasks are naturally unseen by the algorithm. To test 3), we conducted experiments by directly transferring a trained DreamShard to a task with a different number of tables and/or devices without fine-tuning.

Our comprehensive analysis shows that DreamShard can generalize across different table combinations and numbers of tables and/or devices, making it desirable for real-world applications.

## F   Additional Results of DreamShard against Baselines

Table 6: Additional results of running time in milliseconds and relative speedups over random placement on DLRM tasks, measured on 4 GPUs.

| Task | | No strategy Random | Human Experts | | | | RL | |
|---|---|---|---|---|---|---|---|---|
| | | | Size-based | Dim-based | Lookup-based | Size-lookup-based | RNN-based | DreamShard |
| DLRM-10 (4) | Train | 14.8±0.3 | 13.0±0.0 (+13.8%) | 12.8±0.0 (+15.6%) | 11.9±0.0 (+24.4%) | 12.0±0.0 (+23.3%) | 13.3±0.3 (+11.3%) | **11.6±0.3 (+27.6%)** |
| | Test | 13.6±0.3 | 13.0±0.0 (+4.6%) | 12.6±0.0 (+7.9%) | 11.1±0.0 (+22.5%) | 9.6±0.0 (+21.4%) | 12.4±0.1 (+9.7%) | **10.9±0.3 (+24.8%)** |
| DLRM-30 (4) | Train | 32.3±0.5 | 31.0±0.0 (+4.2%) | 28.8±0.1 (+12.2%) | 26.1±0.0 (+23.8%) | 26.2±0.0 (+23.3%) | 30.7±0.8 (+5.2%) | **25.4±0.3 (+27.2%)** |
| | Test | 31.8±0.2 | 30.3±0.0 (+5.0%) | 28.4±0.1 (+12.0%) | 25.4±0.0 (+25.2%) | 25.5±0.0 (+24.7%) | 29.7±0.5 (+7.1%) | **24.6±0.2 (+29.3%)** |
| DLRM-50 (4) | Train | 49.8±0.6 | 49.7±0.0 (+0.2%) | 46.5±0.0 (+7.1%) | 41.2±0.0 (+20.9%) | 41.7±0.1 (+19.4%) | 48.2±1.2 (+3.3%) | **40.4±0.5 (+23.3%)** |
| | Test | 49.8±0.3 | 49.8±0.0 (0.0%) | 45.8±0.1 (+8.7%) | 41.3±0.0 (+20.6%) | 41.4±0.0 (+20.3%) | 48.1±1.2 (+3.5%) | **40.4±0.6 (+23.3%)** |
| DLRM-70 (4) | Train | 66.3±1.0 | 67.8±0.1 (-2.2%) | 63.1±0.0 (+5.1%) | 56.6±0.1 (+17.1%) | 57.5±0.1 (+15.3%) | 70.8±13.2 (-6.4%) | **55.2±0.4 (+20.1%)** |
| | Test | 66.7±0.7 | 69.4±0.1 (-3.9%) | 61.9±0.2 (+7.8%) | 56.5±0.0 (+18.1%) | 57.2±0.0 (+16.6%) | 71.8±15.3 (-7.1%) | **55.2±0.8 (+20.8%)** |
| DLRM-90 (4) | Train | 83.0±1.5 | 82.9±0.0 (+0.1%) | 77.9±0.3 (+6.5%) | 73.1±0.0 (+13.5%) | 73.5±0.0 (+12.9%) | 92.4±13.3 (-10.2%) | **70.0±0.4 (+18.6%)** |
| | Test | 82.3±1.4 | 87.2±0.2 (-5.6%) | 77.9±0.4 (+5.6%) | 71.8±0.2 (+14.6%) | 72.3±0.2 (+13.8%) | 92.9±15.6 (-11.4%) | **69.4±0.7 (+18.6%)** |

Table 7: Running time in milliseconds and relative speedups over random placement on DLRM tasks, measured on 2 GPUs. We observe that DreamShard is comparable with human experts (slightly better than the size-lookup-based method). A possible reason is that these tasks are relatively simple so the expert placements are already near-optimal. Nevertheless, DreamShard still shows strong performance for all the tasks and achieves the best performance in 7 out of 10 tasks.

| Task | | No strategy Random | Human Experts | | | | RL | |
|---|---|---|---|---|---|---|---|---|
| | | | Size-based | Dim-based | Lookup-based | Size-lookup-based | RNN-based | DreamShard |
| DLRM-10 (2) | Train | 17.9±0.2 | 16.4±0.0 (+9.1%) | 16.5±0.0 (+8.5%) | 14.8±0.0 (+20.9%) | **14.7±0.0 (+21.8%)** | 17.0±0.2 (+5.3%) | 15.1±0.3 (+18.5%) |
| | Test | 16.5±0.4 | 16.0±0.1 (+3.1%) | 16.0±0.0 (+3.1%) | 13.9±0.0 (+18.7%) | **13.7±0.1 (+20.4%)** | 16.0±0.2 (+3.1%) | 13.9±0.2 (+18.7%) |
| DLRM-20 (2) | Train | 31.6±0.6 | 30.8±0.0 (+2.6%) | 30.6±0.0 (+3.3%) | 27.4±0.0 (+15.3%) | 27.3±0.0 (+15.8%) | 30.6±0.2 (+3.3%) | **27.1±0.2 (+16.6%)** |
| | Test | 29.9±0.4 | 29.3±0.0 (+2.0%) | 28.8±0.0 (+3.8%) | 26.3±0.0 (+13.7%) | 26.0±0.0 (+15.0%) | 28.8±0.2 (+3.8%) | **25.8±0.2 (+15.9%)** |
| DLRM-30 (2) | Train | 44.6±0.6 | 43.4±0.0 (+2.8%) | 43.0±0.0 (+3.7%) | 39.5±0.0 (+12.9%) | **39.3±0.0 (+13.5%)** | 43.1±0.5 (+3.5%) | **39.3±0.3 (+13.5%)** |
| | Test | 43.7±0.4 | 42.6±0.1 (+2.6%) | 42.1±0.0 (+3.8%) | 38.9±0.1 (+12.3%) | **38.5±0.0 (+13.5%)** | 42.4±0.1 (+3.1%) | 38.6±0.4 (+13.2%) |
| DLRM-40 (2) | Train | 58.7±0.6 | 57.1±0.1 (+2.8%) | 56.2±0.1 (+4.4%) | 53.0±0.0 (+10.8%) | 52.5±0.0 (+11.8%) | 57.5±0.7 (+2.1%) | **52.3±0.3 (+12.2%)** |
| | Test | 58.6±0.7 | 56.9±0.0 (+3.0%) | 56.9±0.0 (+3.0%) | 52.5±0.0 (+11.6%) | 52.4±0.0 (+11.8%) | 56.5±0.4 (+3.7%) | **51.9±0.1 (+12.9%)** |
| DLRM-50 (2) | Train | 72.2±1.2 | 71.2±0.0 (+1.4%) | 70.0±0.0 (+3.1%) | 66.0±0.0 (+9.4%) | **65.5±0.0 (+10.2%)** | 71.5±0.4 (+1.0%) | **65.5±0.2 (+10.2%)** |
| | Test | 72.7±0.6 | 70.6±0.0 (+3.0%) | 70.7±0.0 (+2.8%) | 65.7±0.0 (+10.7%) | 65.6±0.0 (+10.8%) | 70.8±0.5 (+2.7%) | **65.5±0.3 (+11.0%)** |

## G   Additional Results of Generalizability

Table 8: Additional results of the generalization performance of DreamShard from source tasks to target tasks w.r.t. to different numbers of tables. In general, DreamShard can transfer to tasks with different numbers of tables with competitive or even better performances.

| Source \ Target | DLRM-20 (4) | DLRM-40 (4) | DLRM-60 (4) | DLRM-80 (4) | DLRM-100 (4) |
|---|---|---|---|---|---|
| DLRM-20 (4) | - | 32.5±0.3 | 47.8±0.2 | 62.8±0.4 | 77.9±0.4 |
| DLRM-40 (4) | 17.6±0.1 | - | 47.8±0.4 | 62.7±0.5 | 78.0±0.5 |
| DLRM-60 (4) | 17.7±0.1 | 32.5±0.2 | - | 63.1±0.4 | 78.2±0.5 |
| DLRM-80 (4) | 17.6±0.1 | 32.4±0.2 | 47.8±0.3 | - | 78.1±0.5 |
| DLRM-100 (4) | 17.7±0.3 | 32.7±0.4 | 48.1±0.6 | 63.2±0.9 | - |
| DreamShard trained on target | 17.6±0.2 | 32.4±0.3 | 47.9±0.7 | 62.7±0.3 | 77.8±0.8 |

Table 9: Additional results of the generalization performance of DreamShard from source tasks with 4 GPUs to target tasks with 2 GPUs w.r.t. to different numbers of tables. In general, DreamShard can transfer to tasks with different numbers of tables and fewer GPUs with competitive or even better performances.

| Source \ Target | DLRM-10 (2) | DLRM-20 (2) | DLRM-30 (2) | DLRM-40 (2) | DLRM-50 (2) |
|---|---|---|---|---|---|
| DLRM-10 (4) | 14.1±0.2 | 26.2±0.3 | 38.7±0.5 | 52.2±0.7 | 65.3±1.2 |
| DLRM-20 (4) | 13.9±0.1 | 25.8±0.1 | 38.1±0.1 | 51.4±0.2 | 64.5±0.1 |
| DLRM-30 (4) | 14.1±0.1 | 26.1±0.2 | 38.5±0.2 | 52.0±0.2 | 65.2±0.2 |
| DLRM-40 (4) | 14.3±0.1 | 26.2±0.1 | 38.6±0.2 | 52.0±0.2 | 65.1±0.2 |
| DLRM-50 (4) | 14.3±0.4 | 26.3±0.3 | 38.6±0.3 | 52.1±0.4 | 65.3±0.6 |
| DreamShard trained on target | 13.9±0.2 | 25.8±0.2 | 38.6±0.4 | 51.9±0.1 | 65.5±0.3 |

Table 10: Additional results of the generalization performance of DreamShard from source tasks with 2 GPUs to target tasks with 4 GPUs w.r.t. to different numbers of tables. In general, DreamShard can transfer to tasks with different numbers of tables and more GPUs with competitive or even better performances.

| Source \ Target | DLRM-10 (4) | DLRM-20 (4) | DLRM-30 (4) | DLRM-40 (4) | DLRM-50 (4) |
|---|---|---|---|---|---|
| DLRM-10 (2) | 10.8±0.3 | 18.3±0.4 | 25.6±0.6 | 33.8±0.7 | 41.7±0.9 |
| DLRM-20 (2) | 10.6±0.1 | 17.8±0.3 | 25.0±0.4 | 32.9±0.4 | 40.7±0.6 |
| DLRM-30 (2) | 10.9±0.3 | 18.0±0.4 | 25.0±0.6 | 32.9±0.7 | 40.7±0.7 |
| DLRM-40 (2) | 10.8±0.1 | 17.8±0.2 | 24.8±0.2 | 32.6±0.3 | 40.2±0.3 |
| DLRM-50 (2) | 10.7±0.1 | 17.6±0.1 | 24.6±0.1 | 32.3±0.2 | 40.0±0.3 |
| DreamShard trained on target | 10.9±0.3 | 17.6±0.2 | 24.6±0.2 | 32.4±0.3 | 40.4±0.6 |

# H   Additional Results of Training Efficiency

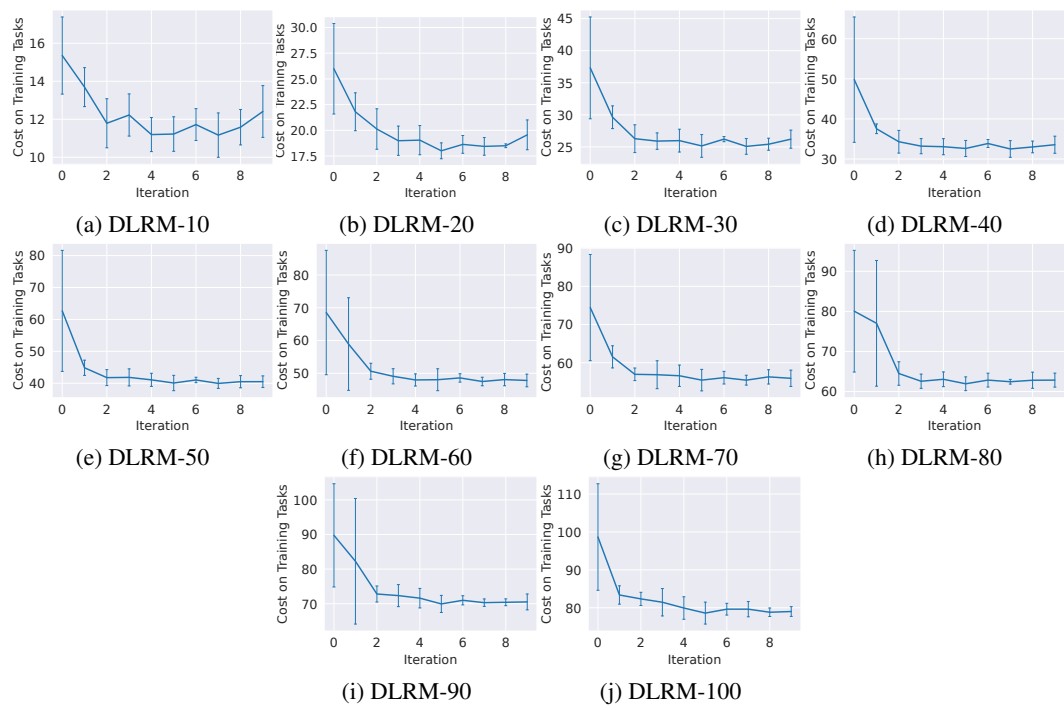

Figure 19: Performance (↓) of DreamShard w.r.t. the numbers of iterations. DreamShard achieves strong performance with very few iterations.

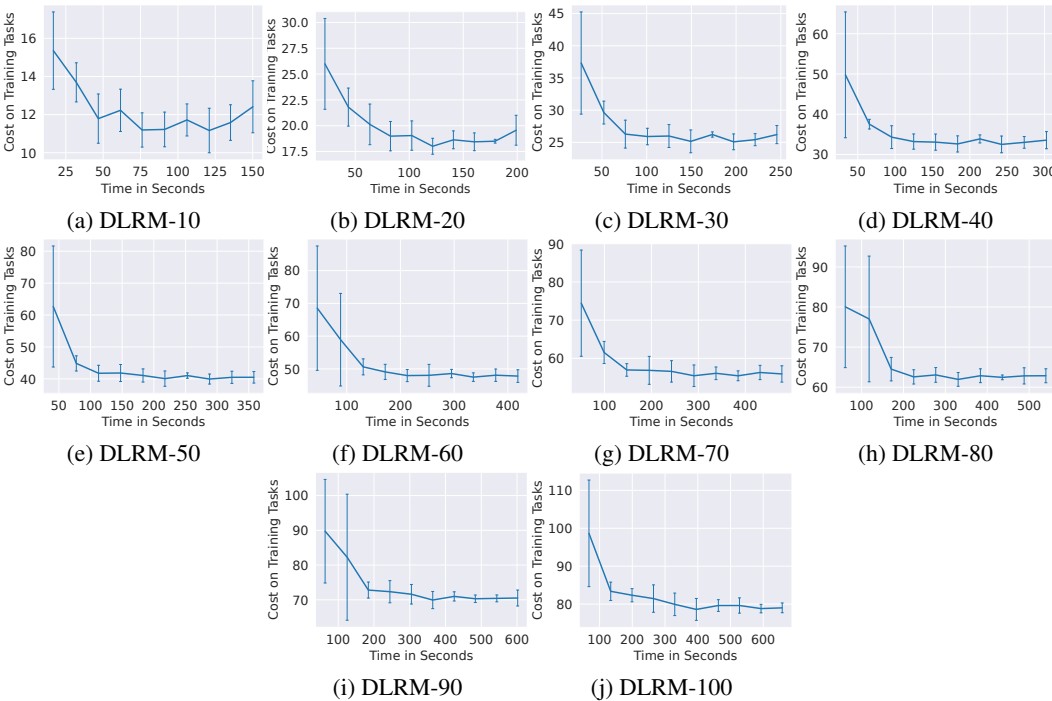

Figure 20: Performance (↓) of DreamShard w.r.t. running time. DreamShard achieves strong performance in a short time.

# I Additional Results of Hyperparameter Study

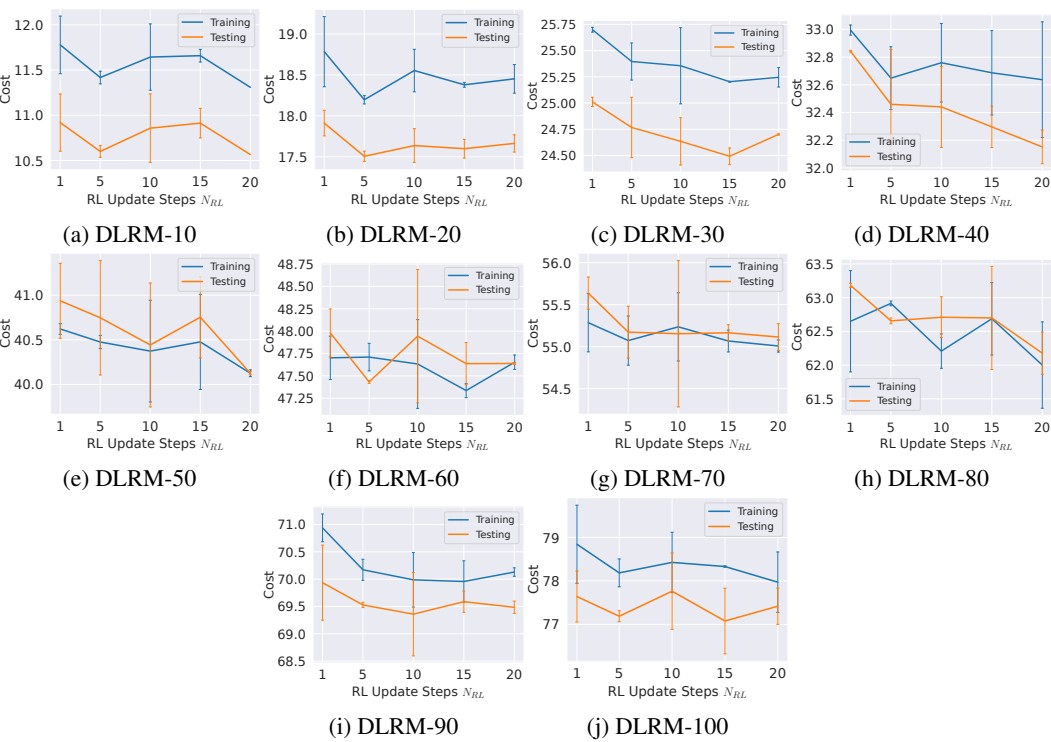

Figure 21: Impact of the number of RL update steps $N_{\mathrm{RL}}$. In general, a too small $N_{\mathrm{RL}}$ will degrade the performance. However, when $N_{\mathrm{RL}} > 10$, increasing $N_{\mathrm{RL}}$ does not lead to a clear benefit, but may cause more computation cost in training RL.

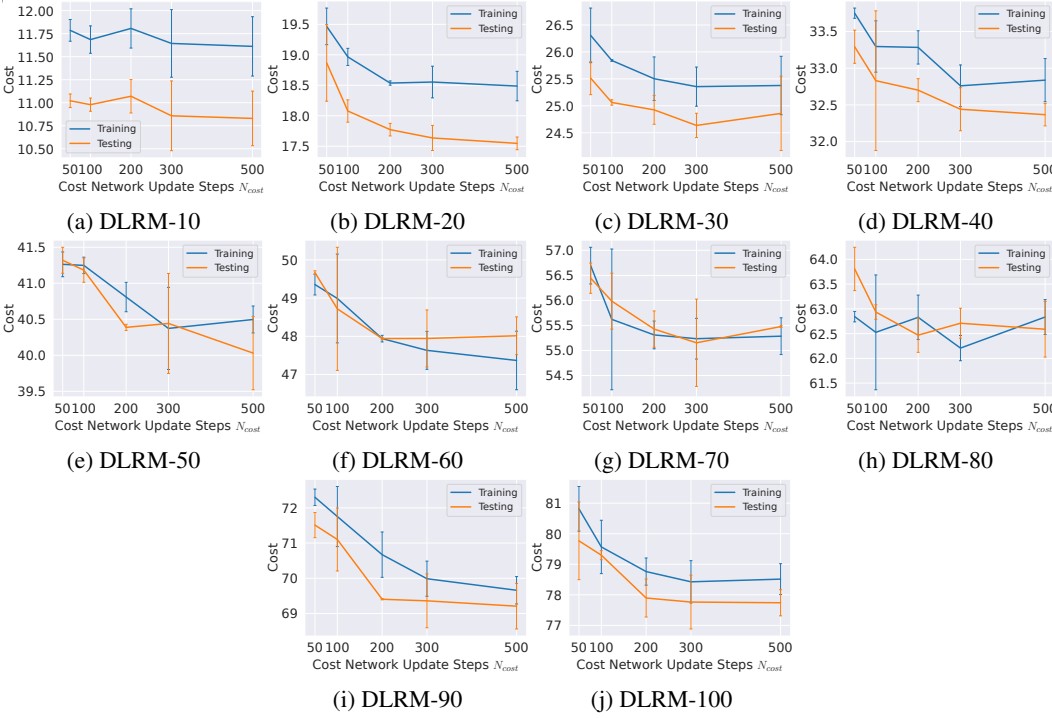

Figure 22: Impact of the cost network update steps $N_{\mathrm{cost}}$. In general, a too small $N_{\mathrm{cost}}$ will degrade the performance. However, when $N_{\mathrm{cost}} > 300$, increasing $N_{\mathrm{cost}}$ does not lead to a clear benefit, but may cause more computation cost in training the cost network.

## J  Additional Results of Ablation Study

Table 11: Ablation Study. The pooling factor and the dimension appear to be the most important table features. The proposed cost features are very effective since w/o cost has poor performances.

| Task | | w/o dim | w/o row | w/o pooling factor | w/o table size | w/o distribution | w/o cost | w/ RNN | DreamShard |
|---|---|---|---|---|---|---|---|---|---|
| DLRM-10 (4) | Train | 11.8±0.2 | **11.5±0.1** | 12.9±0.2 | 11.5±0.0 | 11.8±0.3 | 13.4±0.6 | 11.7±0.1 | 11.6±0.3 |
| | Test | 10.9±0.0 | **10.7±0.0** | 12.4±0.2 | **10.7±0.0** | 11.0±0.4 | 12.4±0.6 | 10.8±0.0 | 10.9±0.3 |
| DLRM-20 (4) | Train | 18.5±0.1 | 18.6±0.3 | 21.6±0.1 | **18.3±0.1** | 18.8±0.2 | 22.0±0.1 | 18.7±0.2 | 18.6±0.2 |
| | Test | 17.7±0.2 | 17.7±0.3 | 20.4±0.2 | **17.5±0.1** | 18.0±0.0 | 20.8±0.3 | 17.8±0.2 | 17.6±0.2 |
| DLRM-30 (4) | Train | **25.3±0.2** | 25.5±0.1 | 29.6±0.2 | 25.2±0.1 | 25.7±0.4 | 29.8±0.5 | 25.2±0.2 | 25.4±0.3 |
| | Test | 24.7±0.3 | 24.9±0.1 | 29.2±0.4 | **24.5±0.1** | 24.9±0.2 | 29.0±0.6 | 24.7±0.3 | 24.6±0.2 |
| DLRM-40 (4) | Train | 33.1±0.8 | 32.9±0.3 | 37.9±0.3 | 33.2±0.4 | 32.9±0.1 | 38.1±0.2 | **32.6±0.2** | 32.8±0.3 |
| | Test | 33.3±0.8 | **32.4±0.2** | 37.9±0.6 | 32.9±0.4 | 32.5±0.2 | 37.2±0.1 | **32.3±0.1** | 32.4±0.3 |
| DLRM-50 (4) | Train | 40.8±0.4 | 40.7±0.1 | 46.3±0.3 | 40.8±0.4 | 40.6±0.2 | 47.5±1.2 | 40.5±0.2 | **40.4±0.5** |
| | Test | 40.9±0.6 | 40.6±0.3 | 47.2±0.1 | 40.6±0.7 | 40.5±0.2 | 46.3±0.1 | 40.5±0.1 | **40.4±0.6** |
| DLRM-60 (4) | Train | 48.5±0.7 | 47.6±0.4 | 54.3±0.2 | 47.8±0.3 | 48.0±0.1 | 53.9±1.3 | **47.5±0.0** | 47.6±0.4 |
| | Test | 48.9±0.5 | **47.7±0.4** | 54.8±0.3 | 48.0±0.3 | 48.1±0.3 | 54.7±0.8 | **47.7±0.1** | 47.9±0.7 |
| DLRM-70 (4) | Train | 56.0±0.5 | **55.2±0.1** | 62.9±0.2 | 55.5±0.2 | 55.3±0.1 | 62.5±0.6 | **55.0±0.1** | 55.2±0.4 |
| | Test | 56.1±0.2 | 55.5±0.2 | 62.8±0.6 | 55.5±0.2 | **55.6±0.0** | 58.3±0.5 | 55.0±0.0 | 55.2±0.8 |
| DLRM-80 (4) | Train | 64.2±0.6 | 62.8±0.1 | 70.3±0.7 | 62.6±0.1 | 62.9±0.1 | 71.4±0.7 | 62.5±0.2 | **62.2±0.2** |
| | Test | 64.2±0.6 | 62.6±0.1 | 71.0±1.1 | 62.9±0.0 | 63.0±0.4 | 71.1±1.4 | **62.1±0.4** | 62.7±0.3 |
| DLRM-90 (4) | Train | 71.8±1.1 | 71.0±0.7 | 79.1±1.0 | 70.7±0.1 | 70.4±0.4 | 79.8±0.9 | 70.8±0.3 | **70.0±0.4** |
| | Test | 70.8±1.2 | 70.3±0.8 | 77.5±0.4 | 69.6±0.3 | 70.1±0.0 | 79.0±1.3 | 70.2±0.1 | **69.4±0.7** |
| DLRM-100 (4) | Train | 79.6±0.3 | 79.1±0.6 | 87.6±0.6 | 78.7±0.2 | 78.9±0.1 | 89.1±2.8 | 78.6±0.6 | **78.4±0.6** |
| | Test | 78.1±0.8 | 78.1±0.5 | 86.2±0.8 | 78.0±0.4 | 78.0±0.2 | 87.7±1.5 | **77.8±0.6** | 77.8±0.8 |

Table 12: Ablation study of the table features on the Prod dataset. We did not use the DLRM dataset because its tables have the same table dimension, which could make the prediction less dependent on the table dimension. We collect a million samples and split 80%/10%/10% as training/validation/testing sets. We fully train a cost network with 100 epochs and report the MSE on the testing set with each individual feature being removed. We find that each table feature contributes to the prediction accuracy, and the most contributing features are table dimension, pooling factor, and distribution features.

| Features | Testing MSE |
|---|---|
| w/o dimension | 13.746 |
| w/o hash size | 0.307 |
| w/o pooling factor | 0.635 |
| w/o table size | 0.305 |
| w/o distribution features | 0.437 |
| All features | **0.303** |

## K  Additional Results on Ultra-Large Industrial Recommendation Model

Table 13: Scalability test. We apply each placement algorithm (excluding the RNN-based method since we find it is very unstable and can not deliver a reasonable performance) to an ultra-large industrial recommendation model, which contains nearly a thousand embedding tables that demand multi-terabyte memory. We run all the placement algorithms on a training cluster with 128 GPUs. We measure the embedding cost and the overall training throughput, which includes embedding lookup, dense computation, data loading, etc. We report the relative improvement over random placement. Since the production model has already been optimized with many iterations, a 5% improvement of training throughput is considered significant. DreamShard shows 27.6% improvement over the strongest baseline.

| Sharding Algorithm | Embedding Cost | Training Throughput Improvement |
|---|---|---|
| Random | 118.3 | 0.00% |
| Size-based | 107.6 (+10.0%) | +4.0% |
| Dim-based | 90.8 (+30.3%) | +13.9% |
| Lookup-based | 102.4 (+15.6%) | +11.9% |
| Size-lookup-based | 109.2 (+8.3%) | +12.8% |
| DreamShard | 61.59 (+92.2%) | +45.3% |

# L  Additional Good/Bad Case Study on Tasks with 50 Tables and 4 GPUs

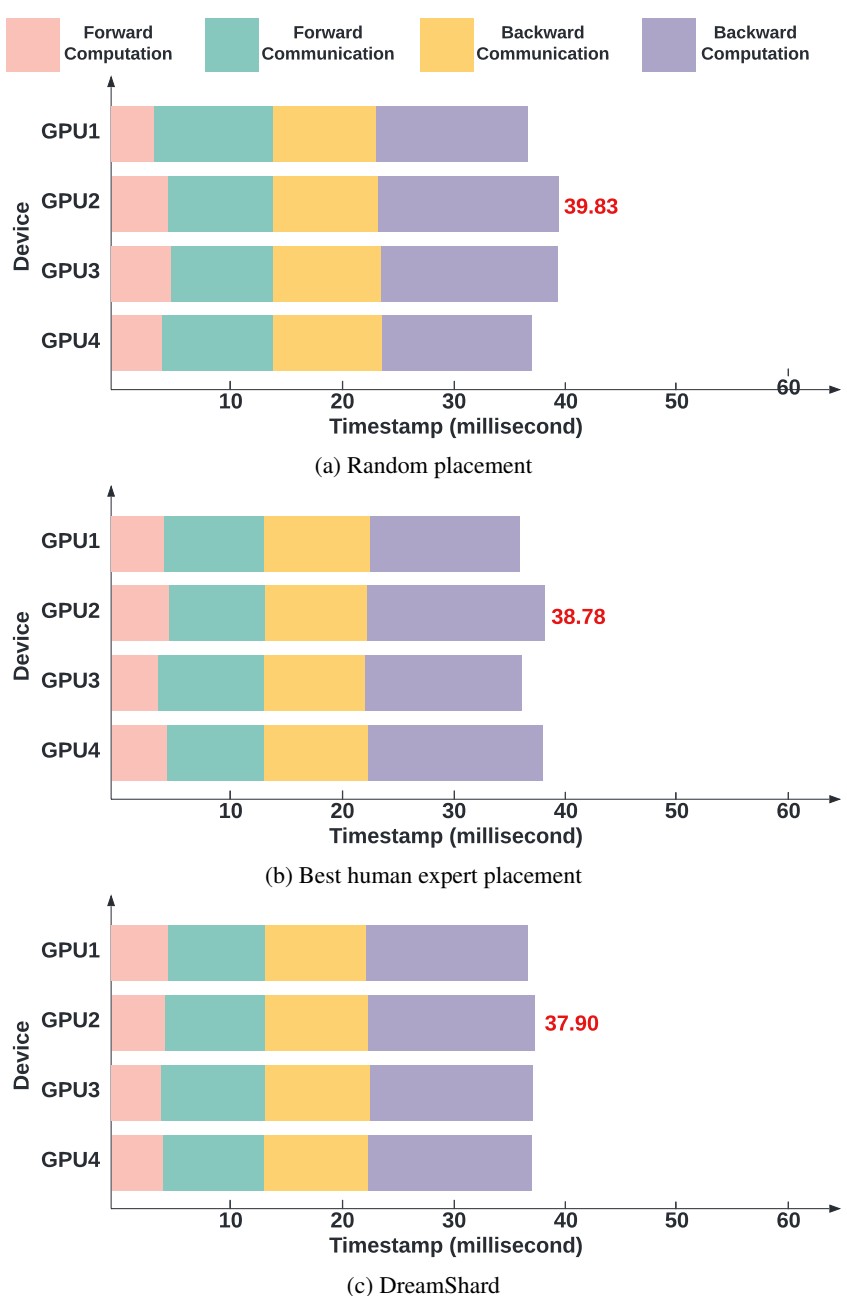

(a) Random placement

(b) Best human expert placement

(c) DreamShard

Figure 23: Good case 1: visualization of DreamShard, the best heuristic algorithm, and random placement on a task of placing 50 tables to 4 GPUs. DreamShard outperforms the baselines with a better balance.

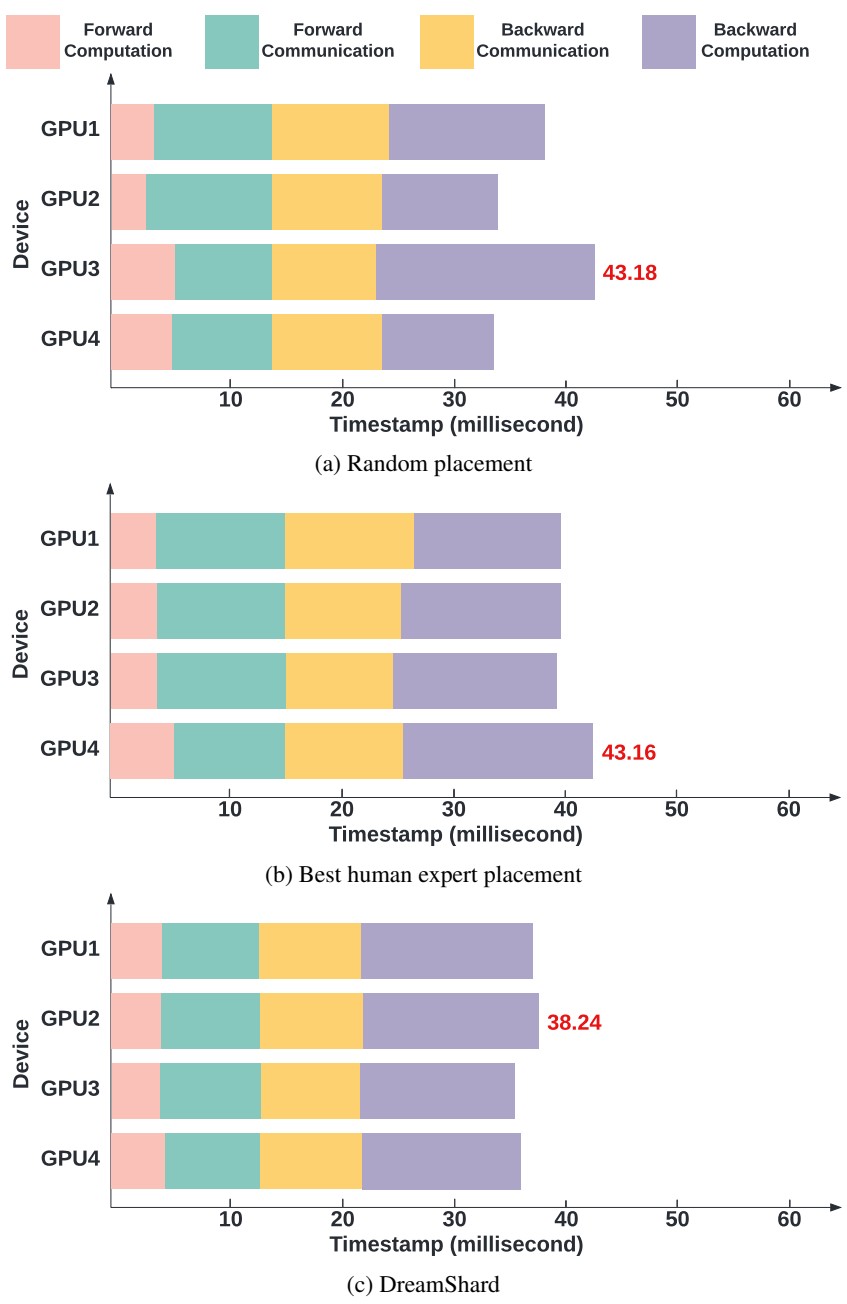

(a) Random placement

(b) Best human expert placement

(c) DreamShard

Figure 24: Good case 2: visualization of DreamShard, the best heuristic algorithm, and random placement on a task of placing 50 tables to 4 GPUs. DreamShard achieves better balance as well as less communication time, leading to significantly lower overall cost.

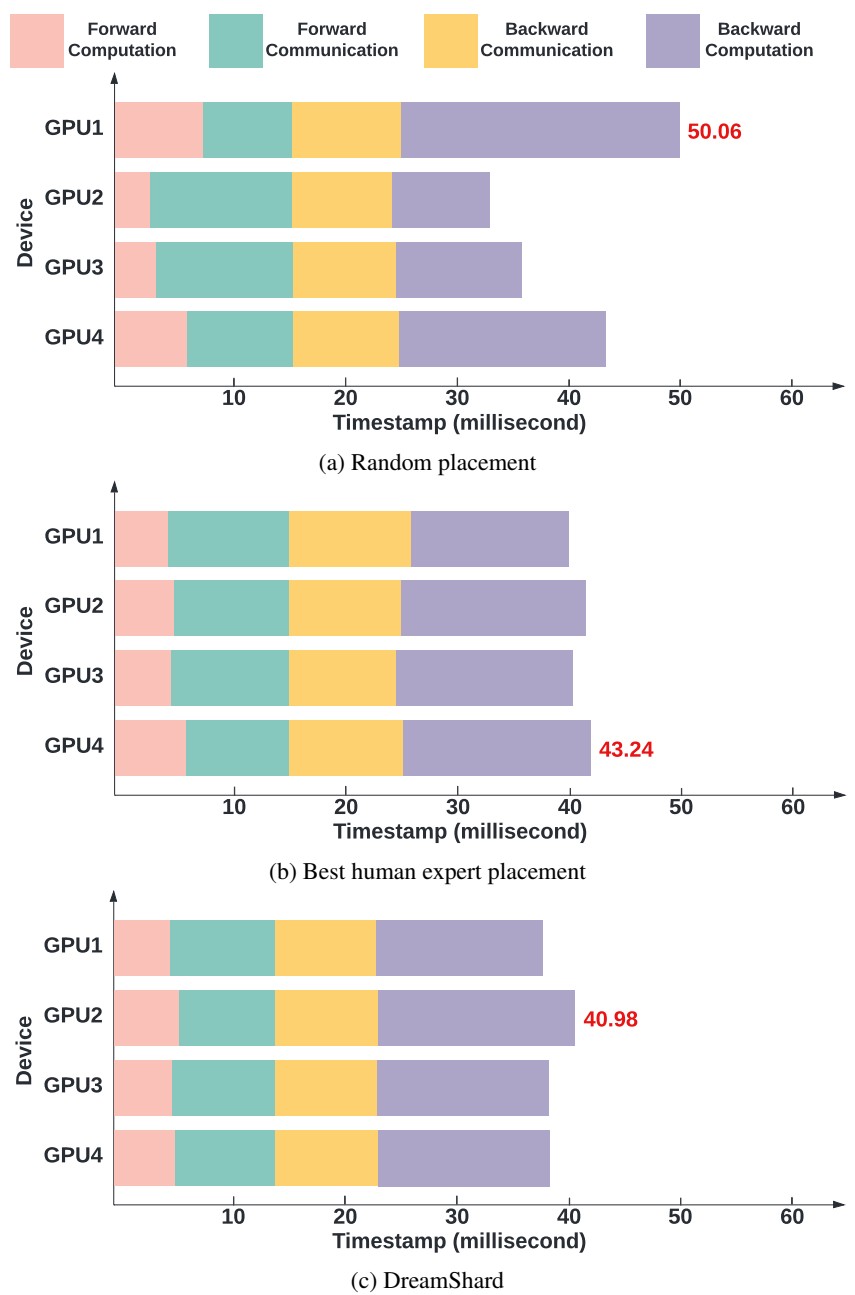

(a) Random placement

(b) Best human expert placement

(c) DreamShard

Figure 25: Good case 3: visualization of DreamShard, the best heuristic algorithm, and random placement on a task of placing 50 tables to 4 GPUs. While DreamShard does not achieve a very good overall balance, the communication time appears to be less than those of the baselines potentially due to a better balance in terms of communication. As such, it still leads to a significant overall speedup.

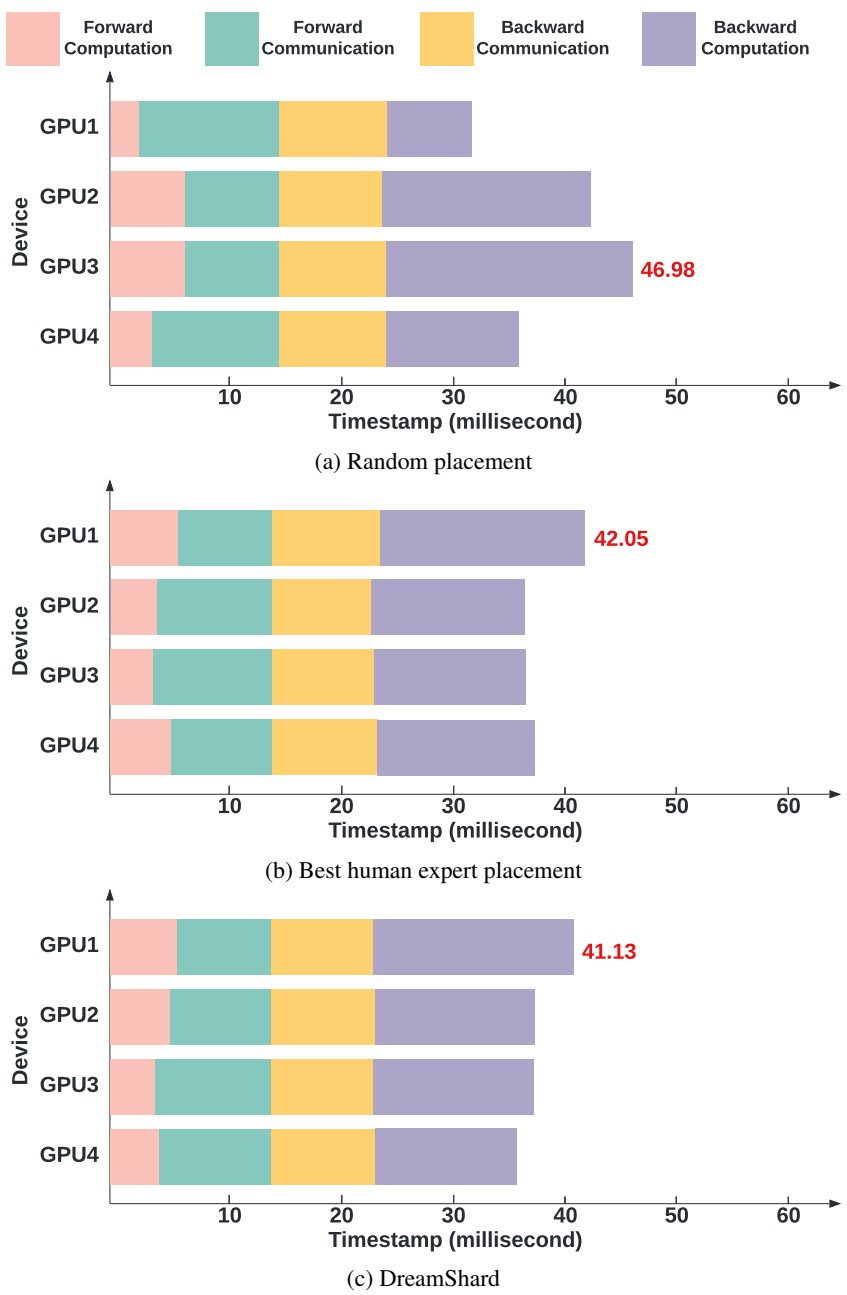

(a) Random placement

(b) Best human expert placement

(c) DreamShard

Figure 26: Bad case 1: visualization of DreamShard, the best heuristic algorithm, and random placement on a task of placing 50 tables to 4 GPUs. The costs of DreamShard are not very balanced. Nevertheless, DreamShard still slightly outperforms the baselines.

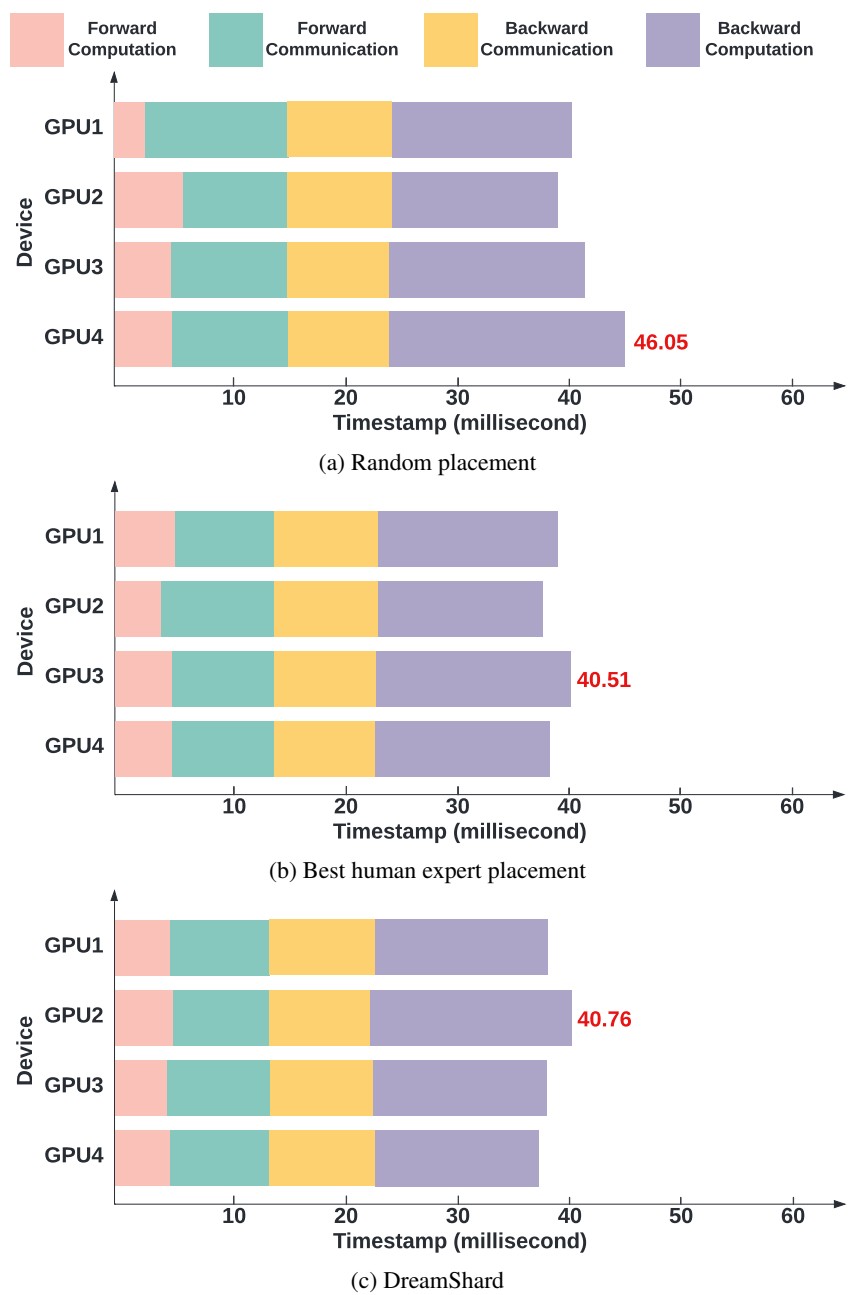

(a) Random placement

(b) Best human expert placement

(c) DreamShard

Figure 27: Bad case 2: visualization of DreamShard, the best heuristic algorithm, and random placement on a task of placing 50 tables to 4 GPUs. The costs of DreamShard are slightly worse than the best heuristic. However, we find that this is actually a very rare case. In most tasks, DreamShard is either significantly better than the best heuristic or has a competitive performance with the heuristic.

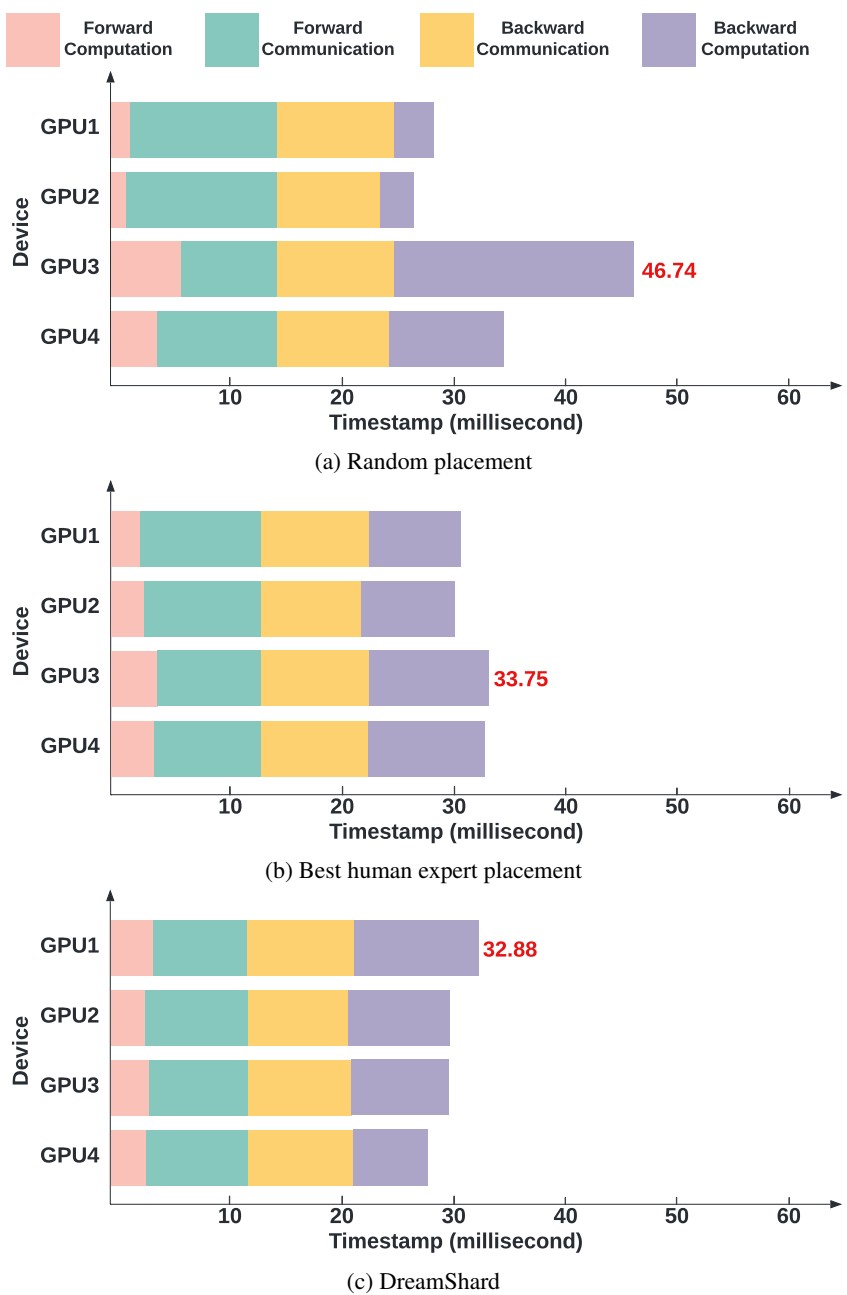

(a) Random placement

(b) Best human expert placement

(c) DreamShard

Figure 28: Bad case 3: visualization of DreamShard, the best heuristic algorithm, and random placement on a task of placing 50 tables to 4 GPUs. While DreamShard achieves better results in the forward pass, it suffers from an imbalance in the backward pass. While it outperforms the best heuristic, it is still very likely to have room for improvement.