# OpenReview forum: "DreamShard: Generalizable Embedding Table Placement for Recommender Systems"
_NeurIPS.cc/2022/Conference — NeurIPS 2022 Accept_

### Official Review · Reviewer_ypFj · 2022-07-09

**Rating:** 6
**Confidence:** 4
**Soundness:** 3 good
**Presentation:** 3 good
**Contribution:** 3 good

**Summary:**

This paper presents DreamShard for embedding table placement in recommender systems to balance the computation and communication costs. The authors formulate the table placement process as an MDP and train a cost network to estimate its states and rewards. Then they update the policy network by interacting with the estimated MDP. The results show speedup over the existing algorithms and good generalizability.


**Questions:**

Pleae see comments above.

**Limitations:**

The authors have discussed the limitations and potential negative societal impact of their work.

**Strengths And Weaknesses:**

### Strengths:
1. The paper is clearly written and very structural. The idea of formulating the table placement process as a MDP is interesting.
2. This paper considers the different speedups across different table combinations and proposes a generalizable embedding table placement problem.
3. This paper addresses the embedding table placement problem with two networks, achieving good performance in speedup and generalization. The experiments are well-designed and thorough.

### Weaknesses:
1. It would be nice to have more results on DLRM with 8GPUs in table 1.
2. Why the author choose 21 table features such as hash size? Can it influence the performance of the model?
3. Lack of comparison with related works, e.g., RecShard.

---

> ### Author Response · Authors · 2022-08-02
> **Thank you for the review**
>
> Thank you for the helpful feedback and comments. Please see our response below.
>
> **1. It would be nice to have more results on DLRM with 8GPUs in table 1.**
>
> We have included 8-GPU results on DLRM in Table 1 (part of 4-GPU results are moved to Appendix due to space limitation). In addition, we have run experiments with 128 GPUs. DreamShard shows consistent improvement in all the settings. **Please see the response to the scalability part above for details**.
>
> **2. Why the author choose 21 table features such as hash size? Can it influence the performance of the model?**
>
> The features are designed based on domain knowledge. In our experiments, we find each of the features can help improve performance.
>
> Table 3 in the paper has compared DreamShard with the variants with each feature being removed. We find that using all the 21 features can lead to the best load balance.
>
> To fully address your concern, we further run an ablation study for the accuracy of the cost model on the Prod dataset. We did not use the DLRM dataset here because its tables have the same table dimension, which could make the prediction less dependent on the table dimension. Whereas, the Prod dataset has more diverse table dimensions, which could better reflect the real feature importance. We collect a million samples and split 80%/10%/10% as training/validation/testing sets. We fully train a cost network with 100 epochs and report the MSE on the testing set with each feature being removed. The results are summarized below (also in Appendix J).
>
> |||
> |--- |--- |
> |Features|Testing MSE|
> |w/o dimension|13.746|
> |w/o hash size|0.307|
> |w/o pooling factor|0.635|
> |w/o table size|0.305|
> |w/o distribution features|0.437|
> |All features|**0.303**|
>
> We observe that all the features contribute to the performance. It is possible that including more features can further improve the performance under our framework. We will investigate this possibility in our future work.
>
> **3. Lack of comparison with related works, e.g., RecShard.**
>
> To the best of our knowledge, RecShard is the only existing work that is designed for embedding table placement. However, it is very challenging to perform a fair comparison for the following reasons.
>
> - RecShard is not open-sourced. It is quite difficult to implement every detail of RecShard to enable a fair comparison.
> - RecShard has a significantly different experimental setting. For example, they use both GPU and CPU memories to place tables. RecShard will identify the hot rows and put them in GPU memory, and put the rest of the rows in CPU memory. In contrast, we mainly focus on GPU devices. In addition, they have only measured computation time. Whereas, DreamShard aims to balance both computation and communication, which aligns better with real scenarios.
>
> Our work is orthogonal to RecShard since they focus more on multi-level memory hierarchy, while we seek to balance both computation and communication by accurately predicting the costs with neural networks and making placement decisions with RL. **We have already contacted and chatted with the authors of RecShard. We have both agreed that it is non-trivial to make such a comparison.** We have also planned to investigate the possibility of combining DreamShard with RecShard to achieve more improvements. Thus, we will leave this comparison to future work.

---

### Official Review · Reviewer_AKvA · 2022-07-10

**Rating:** 6
**Confidence:** 2
**Soundness:** 2 fair
**Presentation:** 3 good
**Contribution:** 2 fair

**Summary:**

This paper focuses on the embedding table placement problem for distributed recommender systems, where the the operation fusion and the generalizability is important and challenging. For solving the problem, the paper formulates the table placement process as an MDP, and proposes an RL framework named as DreamShard which consists of a cost network and a policy network. Extensive experiments show that DreamShard performs well.


**Questions:**

1.  Does the MDP formulation is usual in the related research?

**Limitations:**

I have  no other specific suggestions about the limitations.

**Strengths And Weaknesses:**

Strengths:
1. The paper is well-written and easy to follow.
2. The embedding table placement problem is practical.

Weaknesses:
1. The introduction of the related work is not enough. I am not sure about the novelty of this paper.

---

> ### Author Response · Authors · 2022-08-02
> **Thank you for the feedback**
>
> We thank the reviewer for the feedback. We summarize the novelties of our work compared with the existing studies below.
>
> - **Problem novelty.** The embedding table placement problem is an important problem that has been rarely studied before (also pointed out by reviewer Lt2a). Prior work on embedding tables mainly focuses on compressing the tables to improve embedding table efficiency, which will lead to precision drops. In contrast, we explored an orthogonal direction by studying how the embedding tables should be placed, which will not impact the precision. We demonstrated that optimizing the placements can lead to significant efficiency gains on both the open-sourced DLRM dataset and our production dataset, which could motivate future exploration in this direction.
>
> - **Technical novelty.** DreamShard introduces two novel ideas to tackle the embedding table placement problem, i.e., training a generalizable cost network to predict the costs (i.e., states and rewards in the MDP) and a generalizable policy network to make decisions based on the cost network, enabling a general, effective, and efficient framework for embedding table placement. From the technical perspective, DreamShard makes the following major novel contributions in the context of two lines of prior work.
>
>   - **Device placement optimization.** The existing device placement techniques can be mainly grouped into RL-based algorithms and cost modeling methods. The former optimizes the placement with trials and errors (**effective but inefficient** since it often requires a large number of trials). The latter builds a cost model to reflect the real performance and adopts offline optimization algorithms (**efficient but less effective** because the cost model could be inaccurate). Whereas, DreamShard connects the efforts of these two research lines in such a way as to jointly train a cost network and a policy network in an end-to-end fashion, being **both effective and efficient**. In the left-hand side of Figure 8, we have shown that introducing the cost network makes the training orders of magnitude more efficient than using RL alone, while being equally effective. In addition, DreamShard can generalize to unseen placement tasks thanks to the cost network, which can not be achieved by prior RL-based placement algorithms. The generalizability of DreamShard makes it **extremely efficient on unseen tasks** since we do not need to re-train the networks as in previous work. In the right-hand side of Figure 8, we have shown that DreamShard can place hundreds of tables for unseen tasks within a second.
>
>   - **Reinforcement learning (RL).** While RL has achieved promising results in various domains, one limitation is that RL is often susceptible to overfitting and may fail to generalize to even a slightly different environment. The generalization of RL is an extremely challenging problem that has attracted increasing research attention recently (e.g., see [1] for a survey). Generalizing a policy for embedding table placement problems is a very challenging problem. The policy not only needs to generalize to unseen states (i.e., unseen tables) but also environments with different episode lengths (different numbers of tables) and different action spaces (different numbers of devices). As a comparison, the existing RL generalization work often only focuses on the environments with the same action space [1].  We have demonstrated that by introducing a cost network and applying some representation reduction techniques, the policy network in DreamShard can **successfully generalize to unseen placement tasks with unseen tables and different numbers of tables/devices with neglectable performance drop**. This can be mainly attributed to the generalizability of the cost network, which inherently encourages the generalizability of the policy network by predicting the state features for unseen tasks. We believe this insight itself is a novel contribution to the broad RL community, and the embedding table placement task could serve as a benchmark for RL generalization in future research.
>
> In the updated paper, we have included additional experiments to show that our novel approach is highly generalizable and scalable. It shows strong performance not only on 2, 4, or 8 GPUs in a single server, but also on a 128 GPU training cluster. **Please see the response to the scalability part above for details**. In addition, our code will be open-sourced to facilitate future research, which is a valuable contribution to both device placement and RL research.
>
> [1] Kirk, Robert, et al. "A survey of generalisation in deep reinforcement learning." arXiv preprint arXiv:2111.09794 (2021).

---

> > ### Author Response · Authors · 2022-08-02
> > **Response to your question**
> >
> > **Does the MDP formulation is usual in the related research?**
> >
> > Our generalizable MDP formulation of embedding table placement is unusual from two perspectives:
> >
> > First, while MDP formulation has been used in device placements, it has not been used for embedding table placements, which pose many unique challenges, e.g., operation fusion, ultra-large tables with multi-terabyte memory, and balancing both computation and communication overheads. DreamShard tackles these challenges by jointly training a cost network and a policy network to optimize the reward.
> >
> > Second, the existing MDP formulation for device placement can not generalize to many unseen scenarios (e.g., they often can  only deal with a specific placement problem or at most a similar problem with the same number of devices). In contrast, our formulation can flexibly accommodate different numbers of tables/devices with the generalizable designs of the policy network and the cost network. Unlike previous studies, **DreamShard can solve a family of embedding placement problems instead of a specific placement problem**, which enables it to be easily deployed in real-world recommender system workloads.

---

### Official Review · Reviewer_Lt2a · 2022-07-12

**Rating:** 7
**Confidence:** 2
**Soundness:** 3 good
**Presentation:** 4 excellent
**Contribution:** 3 good

**Summary:**

The paper proposes a RL-based approach for embedding table placement on multiple devices (mostly GPUs), which outperforms human expert and RNN-based baselines.

**Questions:**

* Could the method scale to larger settings? Currently 4 gpus are being used.
* In a real world setting, the vocab distribution constantly changes (e.g. new popular items) which may affect the load balance. Could the method adapt to this dynamic setting?




**Limitations:**

no.

**Strengths And Weaknesses:**

Strengths
* The problem is important and rarely exploreded before
* The proposed method seems reasonable to me, with lots of reasonable heuristic baselines
* The writing is clear and the paper is easy to follow

---

> ### Author Response · Authors · 2022-08-02
> **Thank you for the comment. We have run additional experiments to answer your questions**
>
> Thank you for the feedback! We have run additional experiments to address your concerns about the scalability and vocab distribution changes.
>
> **1. Could the method scale to larger settings? Currently 4 gpus are being used.**
>
> We have added 8-GPU experiments on the DLRM dataset. In addition, we have extended DreamShard to an ultra-large production recommendation model with nearly a thousand embedding tables that demand multi-terabyte memory, which is trained using a training cluster with 128 GPUs. **Please see the response to the scalability part above for details.** DreamShard shows consistent advantages over the baselines. It also leads to significant improvements in training throughput on our production recommendation models.
>
> **2. In a real world setting, the vocab distribution constantly changes (e.g. new popular items) which may affect the load balance. Could the method adapt to this dynamic setting?**
>
> To test whether DreamShard can accommodate vocab distribution changes, in our production model, we train DreamShard on the data collected from an earlier date and apply it to the data collected one month later. We find that DreamShard shows significant improvements over the baselines even when the vocab distribution changes. **Please see the response to the scalability part above for details.** We note that we are not able to do such experiments on the DLRM dataset since it does not provide the data on different dates.

---

### Official Review · Reviewer_a7Cx · 2022-07-12

**Rating:** 7
**Confidence:** 2
**Soundness:** 3 good
**Presentation:** 3 good
**Contribution:** 3 good

**Summary:**

This paper studies the problem of placing embedding tables of a recommender system model in a distributed training environment. How the embedding tables are placed can have significant impact on latency. Embedding lookup consists of 4 stages - In the forward pass, the embeddings are calculated and communicated of dense vector to target devices, while in the backward pass, the gradients are communicated to the device with the embedding table and the gradients are then calculated for the embedding table. The tables can easily lead to imbalances if not placed correctly and thus longer latencies during training. Thus, given a set of embedding tables and devices, identifying the best partitioning strategy is important to minimize costs. However, device placement is a NP-hard combinatorial problem. Further,  operation fusion and generalizability requirement further complicate the problem.

This paper proposes DreamShard, a Reinforcement Learning based approach for embedding table placement. DreamShard solves the problem of operation fusion and generalizability using two key ideas - (1) Learning a cost network to directly predict the cost of fused networks (2) Training a policy network that interacts with an estimated Markov decision process (MDP) without real GPU execution.

**Questions:**

Questions -

1. Can a table be divided and be placed on > 1 devices in this method? Are there scenarios where a table does not fit in a single device?
2. Can the authors also discuss results comparing a simple greedy based approach?
3. Is the shared feature extraction MLP and sum reduction in the cost network and policy network sharing weights?
4. Given the placement of the tables is decided by the RL model and can vary across samples during training, how is operator fusion carried out? Is there a compiler pass after placement to determine what operations on a device can be fused?


**Ethics Review Area:**

["Responsible Research Practice (e.g., IRB, documentation, research ethics)"]

**Limitations:**

Have the authors adequately addressed the limitations and potential negative societal impact of their work? Yes

**Strengths And Weaknesses:**

Strengths

1. The results of the paper are competitive with recent state of the art methods and evaluated on strong baselines
2. The idea to capture cost of communication and operator fusion using a learned network to generalize to unseen tables, different number of devices etc can help provide a practical solution to this problem


Weakness -
None that I can think of.

---

> ### Author Response · Authors · 2022-08-02
> **Thank you for the feedback**
>
> We thank the reviewer for the comments and feedback! Please see the response to your questions below.
>
> **1. Can a table be divided and be placed on > 1 devices in this method? Are there scenarios where a table does not fit in a single device?**
>
> It is possible that some tables can be extremely large and can not fit in the memory of a single GPU. In this case, we will split the over-sized tables (e.g., splitting them column-wise or row-wise). However, splitting tables will often introduce extra costs due to the batching of the embedding operation. For example, suppose there is a table with a dimension of 512 and an operation kernel time of 20 milliseconds (ms). If we split it in half column-wise, then each table will have a dimension of 256. However, the operation kernel time of each table after splitting will often be much larger than 20 / 2 = 10 ms (e.g.15 ms could be a possible value). Thus, the sum of the costs of the two tables will be larger than 20 ms, which introduces extra costs. This is because the operation can do better optimization with batching before splitting.
>
> Thus, in real cases, we often do minimal splitting (or pre-sharding). That is, we only split the tables when the tables can not fit in a single device. Then we apply different placement algorithms to the pre-sharded tables. As an example, we have extended DreamShard to a production recommendation model, where pre-sharding is performed before DreamShard is applied (**please see the response to the scalability part above for more details**).
>
> Table splitting is another interesting direction that is orthogonal to our work. It is a very challenging problem that requires a tradeoff between load balance (splitting tables to smaller pieces makes it easier to achieve load balance) and minimizing total cost (performing too many splitting steps may lead to significantly more overall costs). We will investigate table splitting strategy (e.g., which tables to split, and whether to perform column-wise, row-wise, or hybrid splitting) in our future work.
>
>
> **2. Can the authors also discuss results comparing a simple greedy based approach?**
>
> Greedy-based approaches simply assign the current table to the device with the lowest cost so far. They are sub-optimal because 1) locally optimal decisions may not necessarily lead to globally optimal solutions, 2) The embedding table placement problem has lots of complicating factors, such as balancing both computation and communication, and reasoning about operation fusion. These are not considered in greedy approaches.
>
> In contrast, DreamShard takes all the table features as inputs and makes the placement decisions in a data-driven manner with RL to optimize the reward (i.e., the total cost), which leads to better solutions than greedy approaches.
>
> **3. Is the shared feature extraction MLP and sum reduction in the cost network and policy network sharing weights?**
>
> We used separated weights in our experiments. We will study whether sharing weights is helpful in our future work.
>
> **4. Given the placement of the tables is decided by the RL model and can vary across samples during training, how is operator fusion carried out? Is there a compiler pass after placement to determine what operations on a device can be fused?**
>
> In each training iteration, we will reconstruct the embedding operation based on the decisions made by RL. There is no compiler pass after placement, but instead, all the tables that are assigned in a device will be fused with [FBGEMM](https://github.com/pytorch/FBGEMM/tree/main/fbgemm_gpu). That is, RL will decide which operations will be fused together.

---

### Author Response · Authors · 2022-08-02
**Response to the Scalability**

We thank all the reviewers for the constructive comments and feedback. A common question raised by the reviewers is whether DreamShard can scale to more GPUs (Reviewers Lt2a and ypFj) and whether it can generalize if vocab distribution changes (Reviewer Lt2a). We have added the following two experiments to answer this question.

**1. 8 GPUs on the DLRM dataset (updated in Table 1).**

We summarize the results in the table below. DreamShard shows consistent advantages over the baselines. We have updated Table 1 with the 8-GPU results and moved half of the 4-GPU results to Appendix F. Note that we used V100 GPUs for the 8-GPU experiments rather than 2080-Ti GPUs used in the 4-GPU experiments since we do not have an 8 2080-Ti GPU server at hand.

|||||||||
|--- |--- |--- |--- |--- |--- |--- |--- |
|Task|Random|Size-based|Dim-based|Lookup-based|Size-lookup-based|RNN-based|DreamShard|
|DLRM-40 (8) train|15.6±0.4|14.1±0.0 (+10.6%)|13.4±0.1 (+16.4%)|**9.8±0.0 (+59.2%)**|9.9±0.0 (+57.6%)|16.2±0.8 (-3.7%)|**9.8±0.6 (+59.2%)**|
|DLRM-40 (8) test|15.2±0.2|14.5±0.0 (+4.8%)|13.2±0.0 (+15.2%)|9.5±0.0 (+60.0%)|9.5±0.0 (+60.0%)|16.0±1.1 (-5.0%)|**9.4±0.5 (+61.7%)**|
|DLRM-80 (8) train|25.0±0.2|24.0±0.0 (+4.2%)|21.7±0.0 (+15.2%)|17.1±0.0 (+46.2%)|17.5±0.0 (+42.9%)|51.4±3.9 (-51.4%)|**16.1±0.3 (+55.3%)**|
|DLRM-80 (8) test|25.2±1.3|25.6±0.5 (-1.6%)|20.8±0.0 (+21.2%)|16.7±0.2 (+50.9%)|16.9±0.1 (+49.1%)|53.4±4.6 (-52.8%)|**16.1±0.4 (+56.5%)**|
|DLRM-120 (8) train|34.0±0.3|32.3±0.0 (+5.3%)|29.8±0.0 (+14.1%)|24.5±0.0 (+38.8%)|25.3±0.0 (+34.4%)|58.6±2.7 (-42.0%)|**23.3±0.2 (+45.9%)**|
|DLRM-120 (8) test|33.5±0.5|35.0±0.0 (-4.3%)|29.2±0.0 (+14.7%)|23.7±0.0 (+41.4%)|24.5±0.0 (+36.7%)|58.7±3.1 (-42.9%)|**22.8±0.2 (+46.9%)**|
|DLRM-160 (8) train|42.8±0.3|41.6±0.0 (+2.9%)|39.0±0.0 (+9.7%)|32.0±0.0 (+33.7%)|32.7±0.0 (+30.9%)|58.3±3.5 (-26.6%)|**30.3±0.2 (+41.3%)**|
|DLRM-160 (8) test|41.1±0.0|42.4±0.0 (-3.1%)|36.4±0.0 (+12.9%)|30.8±0.0 (+33.4%)|31.6±0.0 (+30.1%)|59.3±5.4 (-30.7%)|**29.6±0.2 (+38.9%)**|
|DLRM-200 (8) train|51.5±1.2|48.2±0.0 (+6.8%)|48.0±0.0 (+7.3%)|38.9±0.0 (+32.4%)|39.9±0.0 (+29.1%)|68.7±2.4 (-25.0%)|**37.2±0.2 (+38.4%)**|
|DLRM-200 (8) test|50.7±0.2|50.8±0.0 (-0.2%)|44.8±0.0 (+13.2%)|38.0±0.0 (+33.4%)|38.6±0.0 (+31.3%)|70.4±2.8 (-28.0%)|**36.4±0.3 (+39.3%)**|


**2. 128 GPUs on the production recommendation model (updated in Appendix K).**

To further test the scalability, we extend DreamShard to an **ultra-large production recommendation model with nearly a thousand embedding tables that demand multi-terabyte memory**. Since some tables are too large and can not fit in a single GPU, we perform a pre-sharding step by splitting the large tables in half column-wise. We test all the placement algorithms using a training cluster with 128 GPUs (the RNN-based method is excluded because we find it is very unstable and can not deliver a reasonable performance). In addition to the embedding cost, we also report the overall training throughput, which includes embedding cost, dense computation, data loading, etc. To answer Reviewer ypFj’s question about vocab distribution changes, we purposely train the networks on the data from a previous date and apply the pre-trained model on the data collected one month later (e.g., we train DreamShard on the data collected in January, and apply it to the data collected in February so that the vocab distribution could have significant changes). We summarize the embedding costs and relative training throughput improvements over random placement in the table below. Note that we have only performed a single run for each of the training throughput experiments due to the difficulty of resource scheduling.

||||
|--- |--- |--- |
|Placement Algorithm|Embedding Cost|Training throughput improvement|
|Random|118.37|0.00%|
|Size-based|107.63 (+10.0%)|+4.0%|
|Dim-based|90.83 (+30.3%)|+13.9%|
|Lookup-based|102.44 (+15.6%)|+11.9%|
|Size-lookup-based|109.27 (+8.3%)|+12.8%|
|DreamShard|**61.59 (+92.2%)**|**+45.3%**|


For the embedding cost, we observe DreamShard is **67.8% better than the strongest baseline**. For the training throughput, DreamShard shows **27.6% improvement over the strongest baseline**. Note that since the production model has already been optimized with many iterations, a 5% improvement of training throughput is considered significant.

---

### Author Response · Authors · 2022-08-08
**Summary of Contributions and Improvements**

We sincerely thank all the reviewers for the support and for taking the time to provide all the feedback to help improve the paper. As we are approaching the end of the rebuttal/discussion period, we would like to highlight the contributions of our work and summarize the improvements we have made.

**We have made the following major contributions:**
- We have studied embedding table placement, an important problem that has been rarely explored in the literature. The prior work mainly focuses on compressing the embedding tables to improve efficiency, which will lead to precision drops. In contrast, embedding table placement is an orthogonal direction that will not impact the precision.
- We have proposed DreamShard, a general framework for embedding table placement. DreamShard trains a generalizable cost network to predict the costs (i.e., states and rewards in the MDP) and a generalizable policy network to make decisions based on the cost network. DreamShard can solve a family of embedding placement problems instead of a specific placement problem, which enables it to be easily deployed in real-world recommender system workloads.
- We have performed extensive experiments on both open-sourced and production datasets, showing that DreamShard is effective, generalizable, efficient, and scalable. We will open-source our code to facilitate future studies in this direction.

**We have run the following experiments to address reviewers’ questions or concerns:**
- We have run the experiment of 8 GPUs on the DLRM dataset, verifying that DreamShard shows consistent advantages (scalability questions raised by Reviewers Lt2a and ypFj).
- We have extended our approach to an ultra-large production recommendation model with nearly a thousand embedding tables that demand multi-terabyte memory. We have shown that DreamShard can significantly boost the training throughput in a training cluster with 128 GPUs (scalability questions raised by Reviewers Lt2a and ypFj, vocab distribution changes question raised by Reviewers Lt2a).
- We have run additional ablation studies for the accuracy of the cost model (feature importance question raised by Reviewer ypFj)

We have also tried our best to answer the questions from all the reviewers (please see the detailed responses below). As we are approaching the end of the discussion period, please let us know if you have further feedback. We are happy to address any questions/concerns in the remainder of the discussion period.

---

### Meta-Review · Area_Chair_bikb · 2022-08-23

**Recommendation:** Accept
**Confidence:** Certain

**Metareview:**

The paper proposes DreamShard, a RL-based framework for placing embedding tables across multiple devices in distributed recommender systems. DreamShard jointly trains a cost model (to predict the cost of communication and operator fusion for new configurations) and a policy network to make placement decisions based on the cost model. This two step design makes the algorithm more efficient than naive RL solutions and end-to-end training leads to better generalization than model-based offline strategies.

All reviewers agree that the paper is well written and proposes a practical solution to an important problem that is not well studied in the literature. Furthermore, the paper has a strong empirical section that compares DreamShard to strong baselines on open-sourced and production datasets, shows good results and conveys a broad picture of many aspects of their method.

Overall this is a very well executed paper proposing an efficient and practical solution to an underexplored problem. I recommend acceptance.

For the camera ready the authors should include the new scaling experiments they performed to address the reviewers comments. I would also recommend integrate some of the clarifications regarding the contributions and distinctions from prior work (comments to Reviewer AKvA) in the paper.  Also, it might be worth including the greedy baseline numbers for some experiments, just to put the performance into perspective.

**Award:**

No

---

### Decision · Program_Chairs · 2022-09-14

Accept